# In vivo imaging of phosphocreatine with artificial neural networks

Lin Chen [1,2], Michael Schär[2], Kannie W.Y. Chan[2,3], Jianpan Huang [3], Zhiliang Wei[1,2], Hanzhang Lu[1,2], Qin Qin[1,2], Robert G. Weiss[2,4], Peter C.M. van Zijl[1,2] & Jiadi Xu [1,2✉]

Phosphocreatine (PCr) plays a vital role in neuron and myocyte energy homeostasis. Currently, there are no routine diagnostic tests to noninvasively map PCr distribution with clinically relevant spatial resolution and scan time. Here, we demonstrate that artificial neural network-based chemical exchange saturation transfer (ANNCEST) can be used to rapidly quantify PCr concentration with robust immunity to commonly seen MRI interferences. High-quality PCr mapping of human skeletal muscle, as well as the information of exchange rate, magnetic field and radio-frequency transmission inhomogeneities, can be obtained within 1.5 min on a 3 T standard MRI scanner using ANNCEST. For further validation, we apply ANNCEST to measure the PCr concentrations in exercised skeletal muscle. The ANNCEST outcomes strongly correlate with those from [31]P magnetic resonance spectroscopy ($R = 0.813$, $p < 0.001$, $t$ test). These results suggest that ANNCEST has potential as a cost-effective and widely available method for measuring PCr and diagnosing related diseases.

[1] F.M. Kirby Research Center for Functional Brain Imaging, Kennedy Krieger Research Institute, Baltimore, MD, USA. [2] Russell H. Morgan Department of Radiology and Radiological Science, The Johns Hopkins University School of Medicine, Baltimore, MD, USA. [3] Department of Biomedical Engineering, City University of Hong Kong, Hong Kong, China. [4] Division of Cardiology, Department of Medicine, Johns Hopkins University School of Medicine, Baltimore, MD, USA. ✉email: xuj@kennedykrieger.org

Phosphocreatine (PCr) is a high-energy phosphate compound that is abundant in muscle and brain and used by creatine kinase isoenzymes to generate adenosine triphosphate from adenosine diphosphate. PCr plays a vital role in cellular energy buffering and energy transport, particularly in tissues with high and fluctuating energy demands, such as skeletal muscle, cardiac muscle, and brain[1]. The measurement of PCr provides a unique way to achieve insight into cellular energetics, and has shown great potential in many areas, such as for evaluating mitochondrial function in vivo[2], and for identifying peripheral arterial disease[3] and heart failure[4]. PCr concentrations are reduced in several neurodegenerative and muscle diseases. To date, phosphorus-31 magnetic resonance spectroscopy ($^{31}$P MRS) has been the established method for noninvasively detecting and quantifying PCr in vivo[2,5,6]. In addition to PCr measurement, $^{31}$P MRS also provides information about pH, inorganic phosphate, and adenosine phosphates (ATP, ADP, and AMP) in tissue. In practice, $^{31}$P MRS is most commonly applied to monitor the time dependencies of pH and PCr variation during exercise and recovery for assessing mitochondria[7,8]. However, the inherent low detection sensitivity of $^{31}$P MRS results in low spatial resolution and long acquisition times that hinder its wide application. In addition, $^{31}$P MRS is not available on most magnetic resonance imaging (MRI) scanners in clinical practice due to additional hardware costs associated with $^{31}$P excitation/detection that are not used in clinical $^1$H MRI scanners, the significant expense of broadband transmitting/receiving hardware, and the impracticality of having to switch coils in a clinical setting. Therefore, currently, there are no routine diagnostic tests to noninvasively quantify or map the distribution of PCr in tissue with clinically relevant spatial resolution and scan time.

Chemical exchange saturation transfer (CEST) is an MRI sensitivity-enhancing approach that exploits the interaction between exchangeable protons in low-concentration molecules and the water protons detected in MRI, which has shown great potential in detecting various metabolites in vivo[9–11]. CEST MRI does not require special system hardware or coils, and thus can be performed on standard clinical MRI scanners. However, its translation from high-field animal studies to clinical practice has been slowed by the lower field strengths used in the clinic (1.5 T and 3 T), at which the frequency shifts between the exchangeable protons and the water resonance, as well as the contrast-to-noise ratio of the CEST signal, are reduced. Currently, no prior studies have yet been performed to experimentally evaluate human PCr mapping on clinical MRI systems. To explore and develop high-quality PCr mapping using CEST for clinical practice at low field strengths, optimization of the PCr CEST acquisition is required, and the quantification method needs to be robust against the inevitable static magnetic field ($B_0$) and radio-frequency transmit field ($B_1$) inhomogeneities, as well as interference from other saturation transfer components in tissues.

Artificial neural networks (ANNs) are increasingly used in many diverse areas[12–14] to successfully extract relevant features from extremely large, annotated data sets, and utilize them to create predictive tools based on patterns hidden inside. Once trained, ANNs can apply the learned knowledge to analyses of other data and/or solve task-specific problems. In this study, we demonstrate that ANNs can be used for CEST quantification, dubbed hereafter as ANNCEST. More specifically, we show that this trained neural network can accurately and simultaneously predict multiple important properties, including metabolite concentration, the exchange rate of the exchangeable protons, and $B_1/B_0$ homogeneity information, with just the simple input of a Z spectrum (water saturation transfer spectrum) for each image volume element (voxel). After first training and validating ANNCEST using numerical simulations and PCr phantom data

at 3 T, we optimize the PCr CEST acquisition to obtain maximum PCr contrast on human skeletal muscle, and again train and apply ANNCEST. We then show the feasibility of applying ANNCEST to simultaneously quantify the PCr concentration of human skeletal muscle, the exchange rate of the guanidinium protons from PCr, and the $B_0$ and $B_1$ maps on a clinical 3 T MRI scanner. As additional validation, the PCr depletion and recovery in exercised human skeletal muscle were detected and quantified by ANNCEST, and the results were compared with those from $^{31}$P 2D MRS. We also discuss the potential applications of ANNCEST as well as its advantages and limitations. The results suggest that the exchangeable guanidinium protons of millimolar concentration PCr can be exploited to detect it via the water signal in MRI with greatly enhanced sensitivity (molar signal) using CEST MRI, and its concentration can be quantified using ANNs.

## Results

**Validation of ANNCEST.** We implemented ANNCEST with a feed-forward neural network as shown in Fig. 1a. The neural network was trained using Z spectra generated by the Bloch–McConnell equations[15] for various concentrations and exchange rates of exchangeable protons at multiple offset frequencies, and for spatially varying $B_1$ and $B_0$. Gaussian white noise was added to the training Z spectra to mimic the real situation (Fig. 1b). Numerical simulations were performed to validate that ANNCEST can accurately and simultaneously predict metabolite concentration, exchange rate of exchangeable protons, and $B_0$ with the simple input of a Z spectrum per voxel. The simulated Z spectra were generated based on a PCr phantom model at 3T containing two CEST peaks at 1.95 ppm and 2.5 ppm as shown in Supplementary Fig. 1d–f. The exchange rate ratio between 1.95 ppm and 2.5 ppm was set to 1:2.19 according to measures obtained in a PCr phantom at 37 °C using an inversion recovery technique on a 17.6 T NMR spectrometer (see Supplementary Section 2 and Supplementary Fig. 2). The number of hidden layers was optimized to avoid the risk of overfitting (Fig. 1c), and the initial findings suggested that seven hidden layers can provide adequate predicted capacity, because the improvement of performance with more hidden layers was less than $5 \times 10^{-4}$. Therefore, seven hidden layers were used in this study unless otherwise specified. The statistical analyses of the training results are given in the Supplementary Section 3. To test the performance of the trained neural network for quantifying new data, we generate new Z spectra pixel by pixel based on the ground-truth maps shown in Fig. 1d–f. To give a comparison to ANNCEST, Bloch equation fitting was also performed. The quantification results and statistical analysis in Fig. 1g–n show that ANNCEST can yield better fidelity to ground truth compared with Bloch equation fitting, especially for the quantification of exchange rate and $B_0$ in regions with low concentration (detailed values are listed in Supplementary Table 1). It should be noted that ANNCEST is completed within 2 s on a personal computer (Intel i5-6300U CPU with 8 G memory), while the Bloch fitting requires around 18 h on a cluster computer with 8 parallel computing (AMD Opteron 6100 8-core CUP with 16 G memory).

The same neural networks were applied to quantify the Z-spectra obtained from PCr phantoms, and the results are shown in Fig. 2. An excellent correlation ($R = 0.9989$) was observed between the ground truth and predicted phantom PCr concentration. The related Bland–Altman analysis of concentration is shown in Fig. 2f. The exchange rates obtained by ANNCEST were consistent with those obtained using an inversion recovery approach ($260 \pm 40$ Hz, mean $\pm$ s.d.) (detailed values are listed in Supplementary Table 2). The predicted $B_0$ map (Fig. 2e) showed a strong correlation ($R = 0.9969$) with that obtained by water

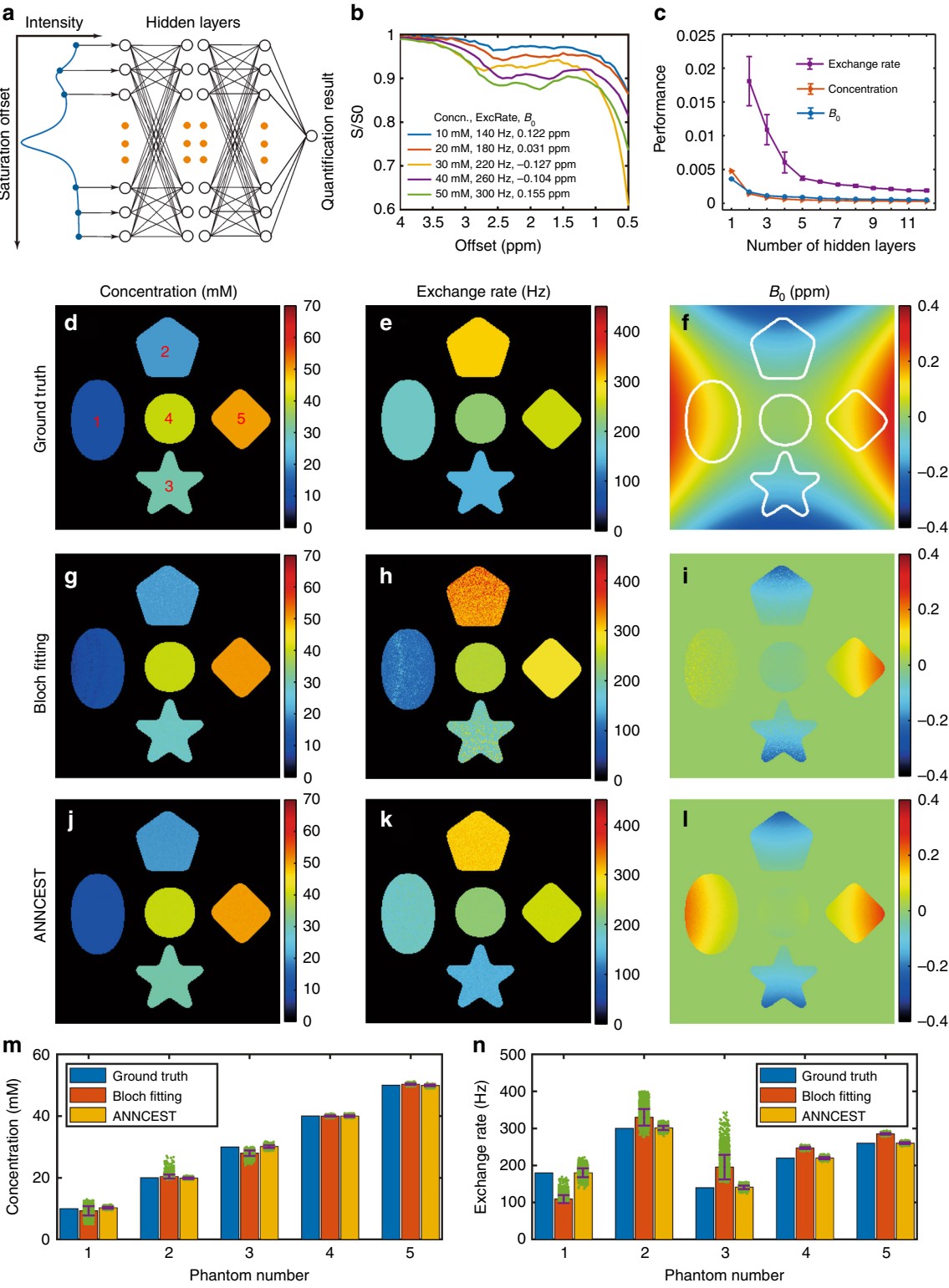

saturation shift referencing (WASSR) MRI[16], as illustrated in Fig. 2h.

**Optimize PCr CEST contrast in vivo on human skeletal muscle at 3 T.** Our previous study on mouse skeletal muscle at 11.7 T showed that PCr has two distinct CEST peaks around 1.95 ppm and 2.5 ppm, and that the 1.95-ppm peak also contains contributions from Cr and protein guanidinium protons[17]. However, the CEST line shapes can be significantly different at lower fields, especially in vivo, where strong magnetization transfer contrast (MTC) exists. Here, CEST experiments with different saturation powers and saturation lengths were performed on human skeletal muscle to experimentally evaluate the PCr contrast on a 3 T clinical system. Due to the presence of MTC, the Z spectra of human skeletal muscle are quite different from those obtained on PCr phantoms (see Supplementary Fig. 1). From the optimization results for saturation powers ranging from 0.2 µT to 0.8 µT (Fig. 3a–d), the Z spectrum obtained with a relatively low

**Fig. 1 Validation of ANNCEST with numerical simulation. a** Diagrammatic representation of the feed-forward artificial neural network used in this study. This network comprises three layers: an input layer, fully connected hidden layers, and an output layer. The input of the neural network is the intensity of the Z spectrum at different frequency offsets, and the outputs are the predicted values of metabolite concentrations, exchange rates of exchangeable protons, and $B_1/B_0$. **b** Representative Z spectra of PCr phantom generated by three-pool Bloch–McConnell equations. Z spectra from 0.5 to 4.0 ppm were sampled with a 50-saturation offset over equal intervals. Gaussian white noise with a standard deviation of 0.35% and $B_0$ inhomogeneity offsets were added to the Z spectra. The saturation power and length were set to 0.6 μT and 10 s, respectively. **c** The performance of neural networks as a function of the number of hidden layers. The error bar was obtained by repeating the neural network training five times. **d–f** The ground truth maps of concentration, exchange rate at 2.5 ppm, and $B_0$ for generating validated Z spectra. The matrix size of maps is 256 × 256. The maps of concentration, exchange rate, and $B_0$ obtained by Bloch equation fitting (**g**, **h**) and ANNCEST (**j–l**), respectively. **m**, **n** Quantified concentrations and exchange rates, respectively. The bar and error bar indicate the mean value and standard deviation across each phantom, respectively ($n$ = 4523, 3655, 2770, 2610, and 2790 pixels for phantom numbers 1–5, respectively).

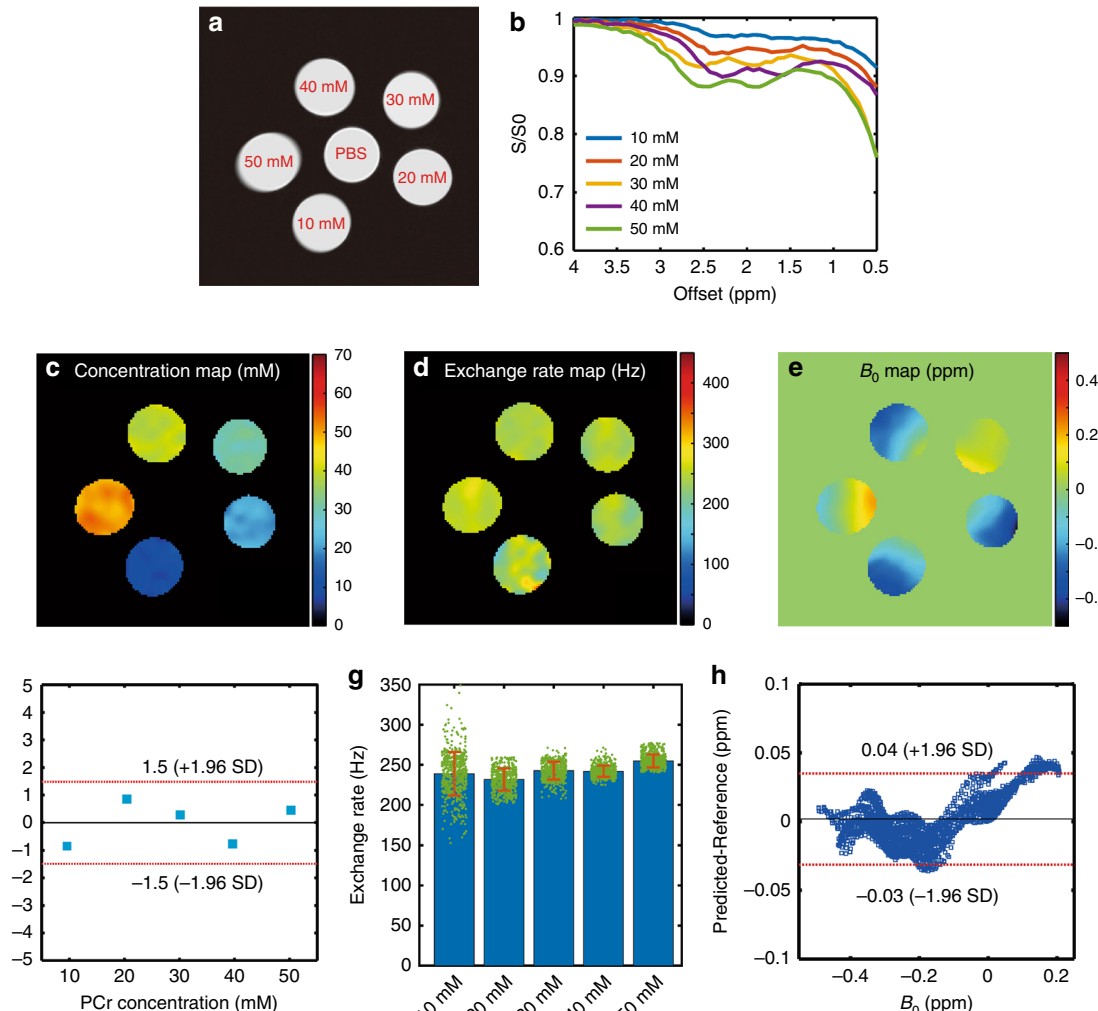

**Fig. 2 Validation of ANNCEST at 3 T (preclinical MRI) on a phantom consisting of test tubes with different concentrations of PCr.** All experiments were performed at 37 °C. **a** The arrangement of the PCr phantoms with different concentrations. **b** Representative Z-spectra extracted from one pixel of each of the PCr phantoms. The ANNCEST-predicted concentration (**c**), exchange rate at 2.5 ppm (**d**), and $B_0$ maps (**e**) for the CEST experiments collected using a 10-s saturation pulse of 0.6 μT. The neural network is identical to the one used in numerical simulation. **f** Bland–Altman plot for the predicted concentration and ground truth. **g** The exchange rate quantification results. The bar and error bar indicate the mean value and standard deviation across each phantom, respectively ($n$ = 579, 473, 508, 587, and 566 voxels for tubes with concentration from 10 mM to 50 mM, respectively). The ground truth of exchange rate (260 ± 40 Hz) was obtained using inversion recovery technique as specified in Supplementary Materials. **h** Bland–Altman plot for the predicted $B_0$ map and referenced $B_0$ map obtained via WASSR method.

saturation power of 0.2 μT exhibits two discernible CEST peaks around 2.5 ppm and 1.95 ppm. With the increase in saturation power, the observed CEST signal (ΔZ) at 2.5 ppm increases at first and then decays after reaching a maximum at 0.6 μT. The CEST peak at 2 ppm is indiscernible from the Z spectra with the saturation power larger than 0.4 μT due to the MTC scale-down effect[18]. The saturation length was optimized with a fixed saturation power of 0.6 μT. From the quantitative analysis shown in Fig. 3i, a CEST protocol with a saturation length of 800 ms yields the maximum ΔZ. Therefore, a saturation power of 0.6 μT and a saturation length of 800 ms were applied in the remaining studies unless otherwise specified.

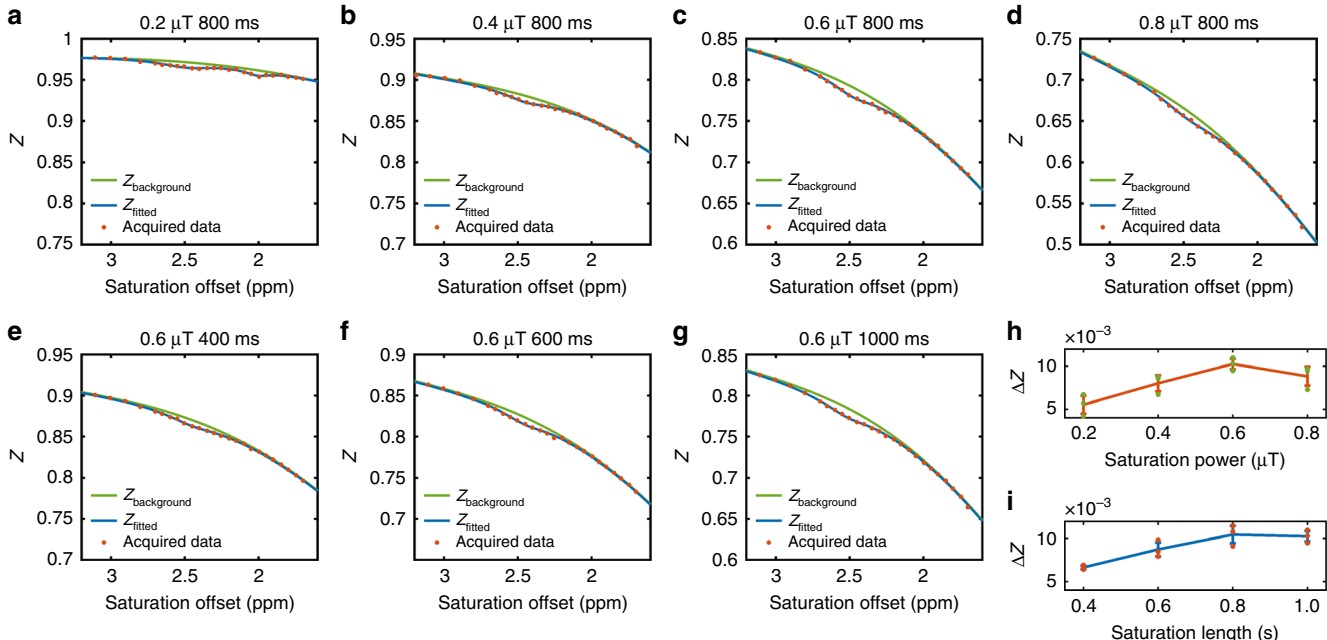

**Fig. 3 Optimization of PCr CEST contrast in human skeletal muscle on a 3 T clinical scanner.** The Z spectra were fitted using a polynomial and Lorentzian line-shape fitting (PLOF) CEST method. The observed CEST signal ($\Delta Z$) indicating the difference between CEST peak and fitted background at 2.5 ppm is used as the evaluation parameter. **a**–**d** Acquired data and fitted results of CEST experiments with fixed saturation length of 800 ms and four saturation powers. **e**, **f** Acquired data and fitted results of CEST experiments with fixed saturation power of 0.6 µT and different saturation lengths. **h**, **i** Quantitative analysis of $\Delta Z$ with different saturation powers and saturation lengths, respectively ($n = 3$ ROIs). Error bars show standard deviations in ROIs.

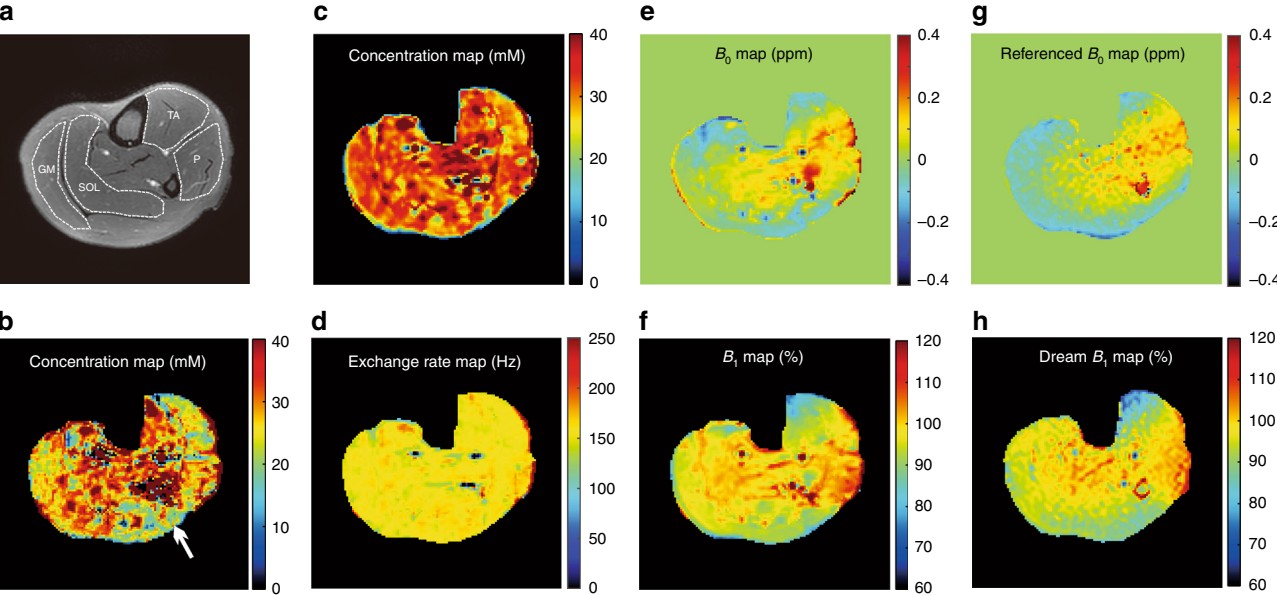

**Fig. 4 PCr mapping on human skeletal muscle using ANNCEST. a** A high-resolution $T_2$ weighted anatomy image with segmentation of gastrocnemius medial (GM), soleus (SOL), tibialis anterior (TA), and peroneus (P). **b** Concentration map obtained using the PLOF method. The PCr concentration (**c**) and exchange rate (**d**) maps together with the $B_0$ (**e**) and $B_1$ (**f**) maps obtained by the ANNCEST method using the CEST images acquired with 0.6 µT saturation power and 800 ms saturation length. The $B_1$ map was normalized by the reference power 0.6 µT = 100%. Reference $B_0$ (**g**) and $B_1$ (**h**) maps of the same slice obtained by the established dual-echo[22] and DREAM[23] methods, respectively.

**High-quality PCr mapping on human skeletal muscle using ANNCEST.** To train the in vivo network, Z spectra of human skeletal muscle at 3 T were generated assuming the experimentally verified situation of a single PCr CEST peak at 2.5 ppm and an additional broad background signal to account for the contributions from MTC and other metabolites that do not show distinct CEST peaks on the Z spectrum. Maps of concentration,

exchange rate, $B_0$ and $B_1$ obtained by applying the newly trained neural network to the in vivo data are shown in Fig. 4c–f. The previously published polynomial and Lorentzian line-shape fitting (PLOF) method was performed for comparison[17,18]. The results showed that the concentration map obtained by PLOF was degraded due to inherent $B_0$ and $B_1$ inhomogeneities as indicated by the white arrow (Fig. 4b), while the concentration map

obtained by ANNCEST shows much better robustness against $B_0$ and $B_1$ inhomogeneities. The quantified PCr concentrations of gastrocnemius medial (GM), soleus (SOL), tibialis anterior (TA) and peroneus (P) over five volunteers were $31.9 \pm 2.0$ mM, $31.7 \pm 3.3$ mM, $30.8 \pm 4.1$ mM, and $30.9 \pm 3.9$ mM, consistent with previous total muscle reports (29–36 mM)[19,20]. The quantified exchange rate ($164 \pm 36.8$ Hz across muscle) is consistent with a previously reported value ($140 \pm 50$ Hz)[21]. The $B_0$ and $B_1$ maps obtained by ANNCEST also show high similarity with those obtained by the established individual methods of dual-echo[22] for $B_0$ and DREAM[23] for $B_1$. Although $B_0$ and $B_1$ showed clear variation in the muscle ($B_0$: $-0.15$–$0.25$ ppm; $B_1$: $0.6$–$0.8$ µT), the ANNCEST method was still able to provide homogeneous concentration and exchange rate maps across the muscle, except for the regions with blood vessels. In order to validate the robustness of ANNCEST against $T_1$ and $T_2$ variations, we applied trained ANNCEST to quantify Z-spectra simulated with different water $T_1$ and $T_2$ values[24] (Supplementary Fig. 10). From Supplementary Fig. 10b, when increasing water $T_1$ over a large range from 1.0 s to 2.0 s, the quantified concentration increased only from $35.03 \pm 1.95$ mM to $37.33 \pm 1.90$ mM, in an approximately linear fashion. The exchange rate obtained by ANNCEST possessed excellent resistance against water $T_1$ variation ($160.7 \pm 2.6$ Hz at $T_1 = 1$ s vs. $159.0 \pm 3.3$ Hz at $T_1 = 2$ s), as shown in Supplementary Fig. 10c. The quantified concentrations and exchange rates as a function of water $T_2$ are shown in Supplementary Fig. 10e, f. These results indicate that ANNCEST still can yield reasonable accuracy when water $T_2$ varies from 15 to 50 ms. We then performed a similar test for sensitivity of ANNCEST to $T_1$ and $T_2$ variation for the PCr protons. The results in Supplementary Section 8 and Supplementary Fig. 11 show that concentrations and exchange rates possess excellent resistance against $T_1$ variation of PCr proton ($34.75 \pm 2.35$ mM and $163.7 \pm 3.5$ Hz at 30 ms vs $36.06 \pm 2.16$ mM and $158.6 \pm 4.9$ Hz at 70 ms). Similarly, ANNCEST still can yield satisfactory accuracy when the PCr proton $T_2$ varies from 15 ms to 25 ms, giving a concentration variation between 34.9 mM and 36.9 mM, while the exchange rate ranges from 162.8 Hz to 159.0 Hz.

The PCr mapping using ANNCEST was further validated by comparison with $^{31}$P 2D MRS measures obtained before and during in-magnet plantar flexion exercise. Subjects underwent the same exercise protocol, once with CEST acquisitions and once with $^{31}$P 2D MRS, in varied order. Shortly after the exercise, the PCr depletion in the gastrocnemius muscles recovered to basal values on the PCr maps obtained by both ANNCEST and $^{31}$P 2D MRS as shown by the representative results in Fig. 5b, c. The PCr recovery time can in principle be obtained from the dynamic PCr maps as demonstrated in Supplementary Section 5. The Z-spectra for a region of interest in the gastrocnemius muscles obtained at two-time points are also plotted and demonstrate the decrease of the PCr CEST peak during the exercise (Fig. 5d, e), namely from 30.38 mM to 15.42 mM. The PCr depletion observed in the different muscle regions is in good agreement with those reported previously in healthy volunteers using $^{31}$P spectroscopy, in which the gastrocnemius muscle showed significantly greater PCr depletion than other muscle groups during plantar flexion exercise[25–27]. PCr concentrations obtained by ANNCEST during exercise agreed very well with those measured by $^{31}$P 2D MRS, as indicated by the correlation ($p < 0.001$, Student's $t$ test) and Bland–Altman analyses between two methods in Fig. 5f, g, respectively.

## Discussion
PCr is an important intracellular high-energy phosphate molecule that is depleted in many neurologic and muscle diseases but one that is rarely quantified or mapped noninvasively in routine clinical practice. Here, we present a technique, named ANN-CEST, to reliably detect, quantify, and image high-quality PCr distribution of human skeletal muscle on a 3 T standard clinical MRI scanner within 1.5 min.

Our previous studies at high magnetic field strength (11.7 T)[17] showed that PCr has two CEST peaks at 1.95 ppm and 2.5 ppm. The same observation was confirmed by another group using 9.4 T and 15.2 T MRI[28]. However, to the best of our knowledge, there has been no published study to map PCr in human skeletal muscle with CEST at clinical magnetic field strength (1.5 T or 3 T). The observed PCr CEST signal at low magnetic field strength is significantly different from that at high fields due to a reduced frequency difference between the exchangeable protons and water protons. From the optimization results shown in Fig. 3, two discernible CEST peaks at 1.95 ppm and 2.5 ppm can be observed for low saturation power (0.2 µT). However, with increased saturation power, the CEST peak at 1.95 ppm becomes indiscernible and only one CEST peak at 2.5 ppm remains. This is fortunate for PCr quantification because the CEST peak at 2.5 ppm is dominated by PCr as validated previously[17,28]. From the PCr phantom experiments shown in Fig. 2, due to the reduced CEST contrast, the exchange rate of 10 mM PCr phantom is more vulnerable to systematic imperfections and exhibits a larger standard deviation compared with the others. Therefore, optimizing CEST sequence to obtain maximum PCr contrast is critical for robust quantification since the PCr contrast in vivo is around 1% (Fig. 3h, i). The quantitation of the metabolites and proteins that form the Z-spectrum is challenging in CEST studies due to the concurrence of the CEST signals from solid-like macromolecules and mobile proteins, as well as the water direct saturation. Many techniques have been proposed for quantification in CEST studies. The most common one is the asymmetry analysis method, i.e., subtracting images acquired at two symmetric offsets with respect to the water resonance, which has been used for quantifying various metabolites[29–31]. Some other methods such as the Lorentzian fitting method[32] and the rotating frame relaxation theory[33,34] have also been proposed for quantification. In most of the CEST quantification methods, the contribution of the CEST signal in the Z-spectrum was usually simplified to a linear function and all contributions were assumed to be superimposed linearly; however, these simplified models usually only work in certain situations, such as with weak saturation powers. CEST quantification is further complicated by the inevitable $B_0$ and/or $B_1$ inhomogeneities, which can shift or distort the in vivo CEST Z-spectra. Usually, $B_1$ and $B_0$ maps need to be collected and included in the CEST quantitation modeling, which not only lengthens the CEST experimental time but also makes the quantification of CEST more challenging. Here, we present the first evidence that ANNs can be used for quantifying in vivo CEST signal. The initial idea of this study was inspired by the work of Bo Zhu et al.[14], who demonstrated that conventional image reconstruction methods can be replaced by ANN-based methods with much-improved performance. Similar to image reconstruction, the CEST quantification is a kind of inverse problem, where the useful information (e.g., concentration and exchange rate) needs to be decoded from the acquired data (i.e., the intensities in a Z-spectrum). The encoding process of CEST MRI can be well described by the Bloch–McConnell equations and a training Z-spectrum can be easily generated with known parameters. However, due to the complexity of the Bloch–McConnell equations, an accurate solution is hard to derive especially with the presence of possible $B_0$ and $B_1$ effects, which means decoding quantitative concentrations and exchange rates from Z-spectra is challenging. Here, ANNCEST provides a new dimension for CEST quantification. A fully connected feed-

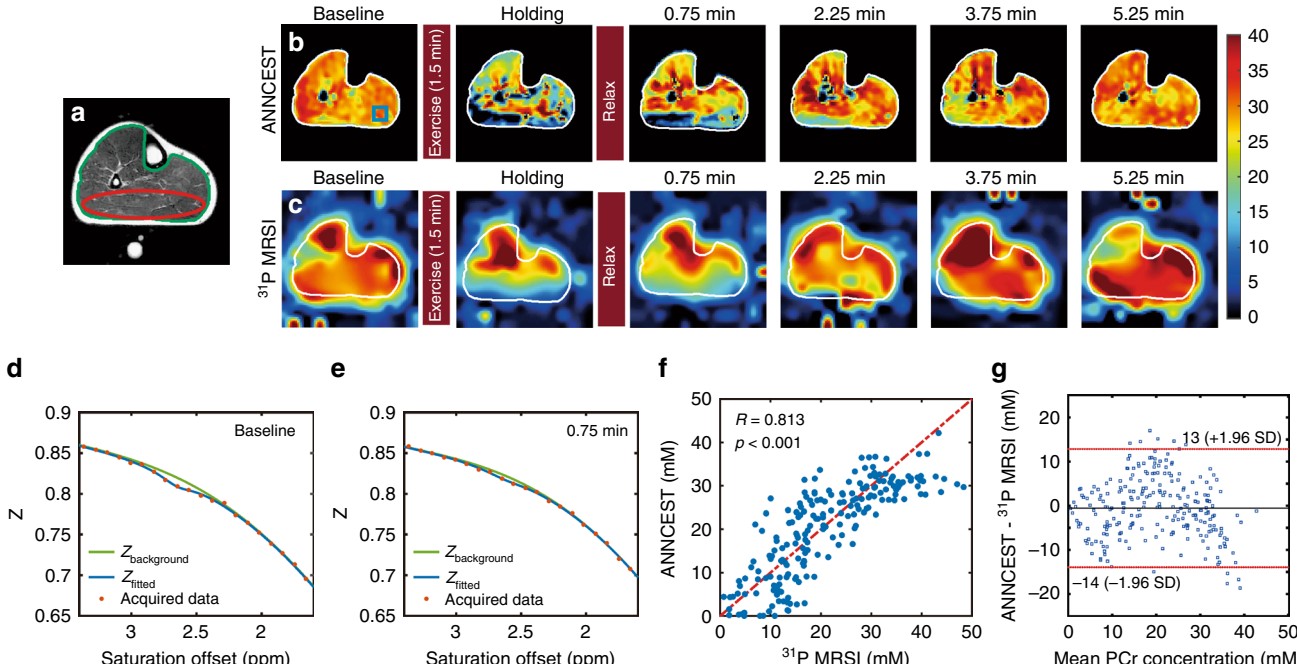

**Fig. 5 Representative PCr maps of human skeletal muscle pre and post in-magnet plantar flexion exercise using $^{31}$P 2D MRS and ANNCEST methods.** **a** $T_2$-weighted anatomy image. PCr concentration maps obtained by **b** ANNCEST and **c** $^{31}$P 2D MRS during the whole exercise process. The Z spectrum at baseline (**d**) and post exercise (**e**) extracted from the blue rectangle region shown in **b**. **f** The correlation analysis of ANNCEST and $^{31}$P 2D MRS from the red ellipse region shown in **a** for all four subjects ($n = 202$ voxels). **g** Bland–Altman plot for the PCr concentrations obtained by ANNCEST and $^{31}$P 2D MRS.

forward neural network with enough neurons in its hidden layer is known to be good at finding patterns behind one-dimensional data[35], which is well suited for CEST quantification. Instead of needing to derive the solution of the Bloch–McConnell equations, we train the ANN with a large annotated dataset to extract the relationship between the Z-spectrum and quantifiable parameters. As shown in Supplementary Section 4, the depth, width, and offset of in vivo PCr peak are related to concentration, exchange rate, and $B_0$ introduced frequency shift, respectively, while $B_1$ variation affects overall Z-spectral background intensity (Supplementary Fig. 5d). These effects can be fully exploited by ANN and applied to simultaneously quantify these parameters (Supplementary Figs. 3 and 4). The ANNCEST approach proposed here provides an additional dimension for CEST quantification. Similar to other artificial intelligence methods, ANNCEST is a data-driven quantification method and its accuracy highly depends on the training data. In this study, the training data were generated using the Bloch–McConnell equations with consideration of $B_0$ and $B_1$ inhomogeneities and noise. The relatively homogeneous $T_1$ and MTC across the human skeletal muscle benefit the generation of training data[24,36,37]. With optimal saturation parameters, training Z-spectra within a limited spectral range (1.3 ppm–3.5 ppm) can be generated using a three-pool Bloch McConnel simulation, namely water protons, PCr guanidinium protons, and background. The background including the contributions from MTC and all other metabolites can be well represented by a single pool (Supplementary Fig. 5e). The discernible PCr guanidinium CEST peak in vivo provides a unique opportunity for ANNCEST to learn the relationships between the Z spectrum and PCr concentration, exchange rate, and $B_0$ and $B_1$ (Supplementary Sections 3 and 4), and to apply the learned knowledge to simultaneously quantify these multiple parameters, as demonstrated in Figs. 4, 5. Previous studies[2,37,38] have shown that the pH, $T_1$, and $T_2$ can change after exercise. We included the effects of pH and T₂ on training data by adopting varying

exchange rates and $T_2$ values. In addition, we tested the performance of ANNCEST using simulated data over a range of water and PCr proton $T_1$ and $T_2$ values (Supplementary Figs. 10 and 11). The results showed that the ANNCEST-determined exchange rates and concentrations are robust over a water $T_1$ range from 1.0 s to 2.0 s and a water $T_2$ range from 15 ms to 50 ms, which are the relevant ones of in vivo ranges[24]. Concentrations were also very insensitive to the PCr proton $T_1$ and $T_2$. ANNCEST can be applied to quantify other metabolites under different situations (see Supplementary Figs. 7, 8 and 9). In the case where the Z-spectrum may be difficult to simulate using Bloch–McConnell equations, the training data can be generated by combining acquired Z-spectra with corresponding quantification results obtained through other gold standard methods. An advantage of CEST MRI for collecting adequate training data is that the size of the training data set is proportional to the number of pixels of the CEST image since each pixel has its own Z spectrum, thus providing sufficient samples and enhancing the power of the resulting data (see Supplementary Section 7).

Though ANNCEST can efficiently exploit the relationship between the Z spectrum and different parameters, the range and frequency offset interval of the Z spectrum and the quantifiable parameters still need to be carefully designed for the successful application of ANNCEST. For the numerical simulations and phantom experiments, the frequency offsets of the Z-spectra ranged from 0.5 to 4 ppm with a total offset number of 50. Within this limited Z-spectral range, $B_1$ and concentration both affect the depth of CEST peak. Therefore, when varying $B_1$ variation (0.5 μT–0.7 μT) in the training, the accuracy of concentration quantification was significantly degraded (the linear regression $R$-value of neural network training dropped from 0.99983 to 0.85054). In this study, due to the relatively small FOV of phantom experiments ($30 \times 30$ mm$^2$), we, therefore, assumed the $B_1$ to be homogeneous across phantoms and a fixed $B_1$ was adopted in the training Z-spectra for numerical simulation and phantom

experiments. This assumption worked well, and high fidelity of PCr concentrations was obtained, as shown in Fig. 2c, f. In phantom experiments where $B_1$ inhomogeneity cannot be neglected, an additional range of Z-spectrum (e.g., the direct saturation of water or more background regions) may be required for accurate quantification of PCr concentration, as well as $B_1$. For the human experiments, successful quantification of $B_1$ and concentration was possible since $B_1$ affects not only the PCr CEST peak but also the MTC-dominant background. An ANN can well exploit this additional information and provide accurate quantification of $B_1$ and concentration as shown in Supplementary Fig. 4g, h. Compared with MRS methods for measuring PCr metabolism of skeletal muscle on clinical scanners such as [13]C MRS[39] and [31]P MRS[2,5,40–42], ANNCEST doesn't require any special MRI transmit or detection coils for heteronuclear or expensive additional system hardware (amplifiers and cabling) and is ready for use on most clinical MRI scanners. In addition, CEST does not require the administration of exogenous contrast agents. With a well-trained neural network, the PCr maps can be calculated rapidly (within a few seconds) from the acquired CEST images, which is much more time-efficient than existing fitting methods, such as Bloch fitting (within a few hours) and PLOF method (within a few minutes), and can provide real-time analysis of a CEST experiment. This is important for the future clinical applications of PCr mapping, where ANNCEST has the potential to address the major target of mapping of the time dependencies of PCr concentration during exercise and recovery. A preliminary result of estimating spatially resolved map of PCr recovery rate constant using ANNCEST is shown in Supplementary Fig. 6 and a recovery time constant of $70.7 \pm 55.4$ s was obtained, which is consistent with that reported in the previous study ($63.1 \pm 25.9$ s)[43]. However, the temporal resolution of PCr ANNCEST in the current study (i.e., 90 s) was too low to capture very detailed dynamic changes. Future possibilities for reducing the scan time of PCr ANNCEST are adopting fewer saturation offsets or utilizing fast CEST sequences[44–46], which needs further study. Finally, as can be seen from Supplementary Fig. 2d, e, the exchange rate depends on both pH and temperature. While we fit out the exchange rate, it is currently not trivial to separate out the effects of pH and temperature and this will be a topic for further study.

In conclusion, as a noninvasive and high-spatial-resolution PCr mapping technique that can be implemented on widely available clinical MRI scanners, PCr ANNCEST has tremendous potential to bring non-contrast metabolic imaging to the clinic whereby identifying, quantifying, and mapping metabolic changes at rest and exercise may guide the diagnosis of many muscle-related, neurologic, and other diseases.

## Methods

**Neural network architecture and training.** A feed-forward neural network composed of one input layer, one output layer, and multiple hidden layers was used for the current study (Fig. 1). The input of the neural network is Z-spectral intensity at different saturation offsets, and the output is the corresponding quantification result. A sigmoid transfer function is used in the hidden layer, and the scales of input and output are normalized to $[-1, 1]$ using a linear function. The weights and biases of the neural network are trained by a scaled conjugate gradient backpropagation algorithm. Some strategies were utilized to avoid overfitting. First, the number of hidden layers was optimized to provide just large enough fitting capacity for the current study. Second, early stopping was applied during the training process. The training data were randomly divided into three sets: training (80%), validation (15%), and test (5%) data. The training data were used to calculate the gradient and update the weights and biases of the neural network. The error on validation data was monitored during the training process. If the validation error increased for 40 iterations, the training was stopped. The test data were used to evaluate the randomness of data division. The test set was designed to compare the performance of different ANN models, and while the error on the test set is not used during training, in practice, it is still useful to monitor. If the error in the test set reaches a minimum at a significantly different

iteration number than the error in the validation set, this might indicate a poor division of the data set. In this study, since the training Z-spectra were generated by randomizing the quantifiable parameters within certain ranges and no other ANN model was adopted, the test set is not critical for the training of neural network. Therefore, we set the portions of the test set and the training set to 5% and 80%, respectively. Finally, a modified performance function, $\gamma \times msw + (1 - \gamma) \times mse$, was used to evaluate the neural network, where $mse$ refers to the mean squared normalized error in training data, $msw$ stands for the mean of the sum of squares of the network weights and biases, and $\gamma$ is the regularization parameter (set to 0.01 in the current study). The application of $msw$ causes the network to have smaller weights and biases, and forces the network response to be smoother and less likely to overfit. Besides the early stopping constraint, the neural network training was stopped once any of the following conditions were met: (1) the calculated gradient was smaller than $10^{-7}$; (2) the mean squared normalized error on validation data was smaller than $10^{-4}$; (3) or the maximum epochs number reached $10^6$. All of the above processing was accomplished on the MATLAB platform (www.mathworks.com, version 9.4.0.813654).

**Z spectra for neural network training.** In this study, training data were generated using the Bloch–McConnell equations with consideration of various imperfect situations, such as $B_0/B_1$ inhomogeneity and noise. Details of the parameters used can be found in Supplementary Table 3 for the phantom and Supplementary Table 4 for the human leg. To mimic the real situation, measurements of water $T_1$ and $T_2$ values were performed for use in this training data. However, it is important to realize that such measurements are not necessary for the ANNCEST applications after the neural network is well trained (See Results Section above). For the numerical simulation and phantom experiments, the frequency offsets of the Z-spectra ranged from 0.5 to 4 ppm with a total offset number of 50. The offsets of PCr CEST peaks were set to 1.95 ppm and 2.5 ppm. The exchange rate ratio between 1.95 ppm and 2.5 ppm was set to 1: 2.19 according to the measurement in a PCr phantom at 37° using a magnetization recovery technique (see Supplementary Section 2). The $T_1$ and $T_2$ of water protons were set to 2.6 s and 1.8 s, respectively, according to the measurements on the phantom. The $T_1$ and $T_2$ values for PCr protons were set to 0.05 s and 0.02 s, respectively. The saturation power and duration were 0.6 μT and 10 s, respectively. The concentration, exchange rate at 1.95 ppm, and $B_0$ inhomogeneity were randomly chosen from the ranges of 5 to 85 mM, 50 to 200 Hz, $-0.4$ to 0.4 ppm, respectively. Gaussian white noise with zero mean value and 0.0015 standard deviations was imposed on the simulated Z spectra. The number of Z-spectra used for neural network training was $10^5$.

For the PCr mapping of human muscle, the frequency offset range of the acquired Z spectra was from 1.3 to 3.5 ppm, with a total offset number of 50. The CEST peak offset was set to 2.5 ppm. The $T_1$ was set to 1.2 s according to the measurement obtained on human skeletal muscle. $T_2$ was varied from 15 to 35 ms. The $T_1$ and $T_2$ values for PCr protons were set to 0.05 s and 0.02 s, respectively. The robustness of ANNCEST against $T_1$ and $T_2$ variations of PCr protons is shown in Supplementary Fig. 11. The saturation power and duration were, respectively, 0.6 μT and 800 ms. The concentration, exchange rate, $B_0$ inhomogeneity, and $B_1$ inhomogeneity were randomly chosen from the ranges of 0 to 100 mM, 80 to 230 Hz, $-0.25$ to 0.25 ppm, and 0.5 to 0.7 μT, respectively. Gaussian white noise with a zero mean value and 0.0035 standard deviations was imposed on the simulated Z-spectra. The MTC-dominated background signal was incorporated as an additional pool with a concentration of 8 M for the exchanging protons and an exchange rate of 30 Hz. The $T_1$ and $T_2$ values for the background signal were set to 1 s and $9.1 \times 10^{-6}$ s, respectively. The lineshape for the background signal was a Super Lorentzian function. The goodness of background fitting using the above parameters is shown in Supplementary Fig. 5e. The number of Z-spectra used for neural network training was $10^5$.

**Numerical simulation.** The ground truth concentration, exchange rate, and $B_0$ maps with a matrix size of $256 \times 256$ are shown in Fig. 1d–f. The Bloch equations were applied to simulate the Z-spectrum for each pixel. In comparison to ANN-CEST, Bloch fitting was performed to quantify the concentration, exchange rates, and $B_0$. The initial values for the ranges of concentration, exchange rate at 1.95 ppm, and $B_0$ were 5 mM (5–85 mM), 50 Hz (50–200 Hz), and 0 ppm ($-0.4$ to 0.4 ppm), respectively. The other parameters were the same as those used for the Z-spectra simulation. The Bloch fitting was performed at a Penguin computer cluster with eight parallel workers (AMD Opteron 6100 8-core CUP and 16 G memory).

**Magnetic resonance imaging of in vitro phantoms at 3 T.** The phantom experiments were performed on a 3 T Bruker Biospec system (Bruker, Ettlingen, Germany). PCr phantoms with different concentrations (i.e., $10 \pm 1$ mM, $20 \pm 1$ mM, $40 \pm 1$ mM, $60 \pm 1$ mM, and $80 \pm 1$ mM) were prepared. Phantoms were prepared in phosphate-buffered saline (PBS) titrated to pH $7.3 \pm 0.1$. All samples were studied in 10 mm glass tubes. A continuous wave saturation module with a saturation power of 0.6 μT and a saturation time of 10 s was applied. Data were acquired using a turbo spin echo (TSE) sequence with TR/TE = 13 s/4.5 ms, TSE factor = 32, slice thickness of 2 mm, acquisition matrix size of $32 \times 32$. Zero filling was applied for the Fourier transform and lead to a reconstructed image with a matrix size of $64 \times 64$, and a resolution of $0.47 \times 0.47$ mm². The CEST experiments

were carried out with only first-order $B_0$ shimming. The $B_0$ map was obtained via the WASSR method[16], in which 21 offsets from $-0.5$ to 0.5 ppm were collected with acquisition parameters and geometry identical to other CEST experiments. The saturation offsets were swept from 0.5 to 4 ppm with an increment of 0.1 ppm. A 0.05 ppm increment was used between 1.5 ppm and 3 ppm to facilitate the detection of CEST peak. The image collected with an offset of 100 ppm was used as the $S_0$ image. The $T_1$ and $T_2$ maps were also collected on the phantom and were applied for the generation of training data. $T_1$ maps were acquired using an inversion recovery sequence with inversion time (TI) from 7 ms to 11.5 s and TR = 15 s, while the $T_2$ map was obtained by a Carr–Purcell–Meiboom–Gill (CPMG) echo train followed by a TSE readout[47] with geometry identical to that of the $T_1$ map measurement. In both $T_1$ and $T_2$ measurements, a slice thickness of 4 mm and a matrix size of $32 \times 32$ were applied. In the $T_2$ CPMG experiment, the inter-echo time was fixed at 5 ms and the total echo time was varied by increasing the number of echoes (40), while block pulses were used for the 90° excitation, and $90^x–180^y–90^x$ composited pulses were applied for the 180° refocusing. $T_1$ values were found to be $2.6 \pm 0.02$ s for all PCr phantoms, while $T_2$ values were found to be $1.8 \pm 0.05$ s.

**Magnetic resonance imaging on human skeletal muscle at 3 T**. The Johns Hopkins Institutional Review Board approved all human studies, with trial registration at ClinicalTrials.gov (NCT04234880). The outcomes of clinial trial are high-resolution PCr and creatine mapping of human skeletal muscle. The inclusion criteria are (1) Subject must be at least 18 years of age. (2) Subject must be willing and able to undergo verbal and written informed consent. (3) Healthy subjects will have no history of cardiovascular or peripheral vascular disease, diabetes, claudication, or difficulty walking. The exclusion criteria are (1) Unable to understand the risks, benefits, and alternatives of participation and give meaningful consent. (2) Contraindications to MRI scan (e.g., implanted metallic objects). (3) Significant cardiovascular (heart failure, significant coronary artery disease, infiltrative or hypertrophic cardiomyopathy, constrictive pericarditis), pulmonary or musculoskeletal, or orthopedic disease that significantly limit exercise capacity. (4) Weight > 350 lbs (inability to fit in the MRI). (5) Cognitive or speech impairments that would limit completion of questionnaires or fatigue reporting. (6) Subjects with rest pain, critical limb ischemia will be excluded for the study. (7) Pregnant women. All subjects gave informed written consent after explanation of the study and protocol. All procedures involving human participants were in accordance with the ethical standards of the institutional and national ethical regulations. The optimization of PCr CEST signal and PCr mapping on resting-state skeletal muscle were performed on a 3 T Philips MRI system equipped with a 16-channel knee coil for radio-frequency transmission and reception. Seven healthy subjects (age $27 \pm 5$ years) were recruited for the PCr optimization and high-quality PCr mapping. The FOV for human skeletal muscle experiment was $160 \times 160$ mm$^2$. High-resolution $T_2$ weighted images were collected for anatomical referencing using TSE sequence with a resolution of $0.63 \times 0.63 \times 5.0$ mm$^3$. A continuous wave saturation module with a saturation power of 0.6 μT and a saturation time of 800 ms was applied for PCr mapping on resting-state human skeletal muscle. Images were acquired using a single-shot TSE sequence with TR = 3.5 s, TE = 3.7 ms, TSE factor = 37, and a resolution of $2.2 \times 2.2 \times 5.0$ mm$^3$. The Z-spectra between 1.3 ppm and 3.5 ppm were sampled together with two $S_0$ images collected at an offset of 100 ppm. Number of offsets acquired for the Z-spectrum was 52 and total scan time was 3 min. $B_0$ maps were obtained using a dual-echo sequence with TR = 10 ms, TE = 4.6 ms, and flip angle of 30°. $B_1$ maps were obtained using the dual refocusing echo acquisition mode (DREAM) technique with a stimulated echo acquisition mode (STEAM) flip angle of 60°[23]. When optimizing the PCr signal in muscle, the extraction of the PCr signal was achieved by the PLOF method[17,18]. The Z-spectrum between 1.6 and 3.5 ppm was utilized for the two-step fitting in the PLOF method, and two regions of the Z-spectrum, i.e., 1.6–2.1 ppm and 2.9–3.5 ppm, were selected for the background fitting. A single Lorentzian function was assumed for the PCr CEST peak at 2.5 ppm. The observed PCr CEST signal $\Delta Z_{PCr}$ was given by taking the difference between the fitted background $(Z_{back}^{ss})$ and the observed $Z$ value at the 2.5 ppm, i.e., $(Z_{back} - Z_{2.5})$.

The CEST experiments and $^{31}$P 2D MRS on exercised human skeletal muscle were performed on a second 3 T Philips MRI system equipped with a 2-channel surface coil for $^1$H imaging and a $^{31}$P excite/receive coil placed beneath the calf muscle for $^{31}$P 2D MRS. Four healthy subjects (age $27 \pm 5$ years) were recruited for in-magnet plantar flexion exercise. The exercise protocol involved plantar flexion exercise with repetitively lifting a 16 lb weight at a rate of 1 Hz for 80 s, and then holding the load for 90 s before stopping all exercise to allow subsequent measures of postexercise PCr recovery[48]. CEST images were acquired using a single-shot TSE sequence with TR = 3 s, TE = 9.3 ms (the shortest TE available), TSE factor = 37, and a resolution of $2.2 \times 2.2 \times 20.0$ mm$^3$. The Z-spectra between 1.3 ppm and 3.5 ppm were sampled together with one $S_0$ image collected at an offset of 100 ppm. Number of offsets acquired for the Z-spectrum was 30 and total scan time was 1.5 min. The $^{31}$P 2D MRS was performed with TR = 1.5 s, TE = 1.44 ms, and a total scan time of 1.5 min. 60 measurements on an $8 \times 8$ k-space were acquired except for the four corners (circular k-space shutter). The k-space data were then reconstructed on a $16 \times 16$ grid leading to a voxel size of $10 \times 10$ mm$^2$. The slice thickness of $^{31}$P 2D MRS was determined by the excitation profile of the $^{31}$P coil, which is in the order of 80 mm.

**Correlation analysis of PCr maps obtained by ANNCEST and $^{31}$P 2D MRS**. A correlation analysis was performed to compare PCr maps obtained by ANNCEST and gold-standard $^{31}$P 2D MRS on resting and exercised human skeletal muscle. To remove the location mismatch in the CEST images caused by motion, image registration was applied to ANNCEST results using the Medical Imaging Registration Toolbox (MIRT). Because $^{31}$P 2D MRS is less sensitive to motion with worse spatial resolution, the ANNCEST results were downsampled from $72 \times 72$ to $16 \times 16$ to match the matrix size of $^{31}$P 2D MRS. The downsampling map was calculated by averaging the intensity within the corresponding patch on high-quality PCr mapping. The PCr maps of baseline, during holding, and 0.75 min of recovery were chosen for comparison. Because the $^{31}$P was placed beneath the skeletal muscle and may lead to poor sensitivity to the upper-half FOV, only the regions close to the coil were chosen for correlation analysis. The pixel-by-pixel correlation analysis was accomplished using Matlab built-in function "corrcoef".

**Reporting summary**. Further information on research design is available in the Nature Research Reporting Summary linked to this article.

## Data availability
The source data underlying Figs. 1d–n, 2c–g, 3a–i, 4b–h, 5b–g are provided as a Source Data file. The other data that support the findings of this study are available from the corresponding author upon reasonable request.

## Code availability
The code used in this study is provided in Supplementary Data 1. We also deposit the code in https://github.com/LinChenMRI/ANNCEST.git.

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

## Acknowledgements

This work was supported by National Institutes of Health grants R01EB015032, P41EB015909, R01HL61912, R01HL63030, R03NS109664, and DOD W81XWH-18-1-0797. The 3 T human MRI equipment in this study was funded by NIH grant 1S10OD021648. We thank Tricia Steinberg for assisting in the in-magnet plantar flexion exercise study.

## Author contributions

L.C. and J.X. designed and performed experiments, processed data, and wrote the paper; P.vZ. and R.W. helped with designing the study, interpreting the data, and paper editing; R.W. and M.S. helped with the $^{31}$P 2D MRS experiments and processing. K.C. and J.H. prepared the phantoms and performed the phantom MRI experiments; M.S. and Q.Q. provided pulse sequence design and technical guidance on the human subject scanning. H.L. and Z.W. helped with human experiments.

## Competing interests

The authors declare no competing interests.
