## [Peer Review File · Nature Communications]

Reviewers' comments:

Reviewer #1 (Remarks to the Author):

It's well known that ³¹P MRS offers quantitative insight into the metabolic processes underpinning muscle function and its pathology. Although resting-muscle measurements of pH, Pi, PCr etc can be informative, the most useful measurements in this regard are time-series of pH and PCr during exercise and recovery protocols. ³¹P MRS has well-known limitations of hardware, signal-to-noise and spatiotemporal resolution (well described in this paper's Introduction) which have limited its clinical and indeed also research application. This paper describes the extremely thorough optimisation and validation of a novel and ingenious approach to PCr measurements in skeletal muscle implementable on clinical scanners, which offers a partial way round these limitations. In metabolic terms, the main thing missing, compared to ³¹P MRS, is pH measurement. The other main limitation is temporal resolution, and I would suggest that this is the next thing to improve.

Against this background, I make only two suggestions about the manuscript:

1. I suggest that a very brief summary of the points I have made in relation to the background and possible use of this technique would usefully set the scene for the less specialist reader.
2. Obviously this would be marginal, given the current time resolution, but is it in any way feasible that the ANNCEST timecourse data in Fig 5a (and for comparison the ³¹P MRSI data in Fig 5b), specifically the recovery-from-exercise timepoints, could be used to calculate a spatially resolved map of PCr recovery rate constant? Such a thing would be interesting and physiologically relevant, being straightforwardly interpretable in terms of spatial distribution of mitochondrial function, and is surely a major way this technique could and should be developed. If (as I suspect) the data are too sparse, temporally, to support even a proof-of-principle calculation at this stage, I think it would be worth saying how this would work and why it would be useful. The point is, of course, that such kinetic measurements probe muscle energy metabolism and its abnormalities in a more effective (in some sense e.g. (sensitive, interpretable) way than resting [PCr] measurements.

Reviewer #2 (Remarks to the Author):

Summary

The authors presented a pipeline that utilizes artificial neural networks (ANN) to quantify Phosphocreatine (PCr) concentration with high immunity to MRI interferences including magnetic field (B0) and radio-frequency transmission (B1) inhomogeneities. The methods have been tested on both phantoms and exercised skeletal muscle. The study documents the ability of the ANNN -based chemical exchange saturation transfer (ANNCEST) for measuring PCr and diagnosing related diseases.

Strength

- The paper introduces a framework for an important application that has a great interest to researchers for medical image analysis society, especially those pursuing in-vivo PCr quantification.
- The study is well-designed and validated
- Most of the literature work references is up-to-date
- The paper is well- written, and results are nicely discussed

Weaknesses

- The core computational algorithm used in the pipeline (ANN) is already published, which is not a problem in itself, but the rationale behind the specific choice should be discussed. Additionally, the scientific contribution of the manuscript, however, has been compensated for by the application itself and method evaluation.
- The online Methods is misplaced after discussion. How many is the training data? Also, 80-15-5 split is the author's choice or a standard procedure? Although the authors adapt the ANN, they still need to add details about their method, optimization, hyper-parameter tuning, etc.
- One more limitation of the study is the data size. Power analysis should be conducted to determine the appropriate data size that can be used to draw the author's conclusions.
- Abstract does not contain any quantitative results. Also, please refrain from using abbreviations from abstract, unless you define them (see e.g., WHM)
- Quantitative results and statistical analysis have been conducted and included, however, comparison with other techniques is missing.

Reviewer #3 (Remarks to the Author):

The robust imaging of PCr on a 3T clinical system in 1.5 minutes is an exciting result that could have significant interest in the medical imaging field. While there are lots of interesting results, the logic

and assumptions of the paper are not fully clear, with specific relevance to the benefit of the neural network approach vs the PLOF approach, lessening enthusiasm.

1. The overall logic of the paper is not fully clear, which is especially confusing since it is trying to combine what could be several papers (PLOF at 3T, ANN in steady-state vs non-steady state or phantoms vs in vivo, ANN validation with MRS, and ANN vs PLOF). It is divided into 5 parts: 1. ANN works on numerical 2-pool phantoms, and works better than PLOF/numerical fitting; 2. PCr generated signal contrast is maximized at a power and duration in vivo at 3T (which is different than that used in the digital or in vitro phantoms); 3. ANN works on phantoms at 3T, but with systematic spatial artifacts (which weren't discussed). 4. ANN and PLOF work in muscle at 3T but ANN is more uniform and maybe better. 5. ANN results match MRS in exercised muscle. I'm not certain this is the division, which is part of the problem. The logical progression should be clearer. For example, the # of pools, experimental parameter ranges (including changing from steady state to non-steady state conditions), and fitting constraints change between the phantoms and the in vivo work. So what is being tested? Is it the general ANN approach? Wouldn't the validity of the approach depend on how valid the assumptions are, which is very different in vivo where, for example, the MTC parameters were unknown. The logic should be clearly stated, and the results and methods should match this logic and follow a clear progression.

2. Were all simulations, training, and measurements (except for a separate exchange rate measure) done at 3T? The legend in figure 1b indicates a 300Hz variation, which would match the stated 0.4 ppm at 17.6T, not 3T. If the phantom work was done at 17.6 T, I don't understand the logic.

3. What are the assumptions and constraints, and what drives these decisions? From my reading, assumptions in the numerical training of the phantoms include 2 pools, the ratio of exchange rates at 2 and 2.5 ppm (only good at a single temperature and pH?), $B_1 = 0.6 \text{ uT}$ with no inhomogeneity, and water T_1 and T_2 values. The in vivo training makes very different choices, and many of the parameters are not listed (e.g. MTC T_2 and lineshape). In both cases, it is unclear what assumptions are made about the PCr T_1 and T_2 , but they must be in the coupled Bloch equations. The training assumes a single T_1 , but doesn't that change with, for example, exercise? Discussion of possible problems and systematic errors is necessary, along with discussion of why these constraints were chosen (e.g. is it impossible to do the fitting without specifying the ratio of exchange rates?).

4. Is there a fair comparison to PLOF approach? Are the assumptions the same? Does PLOF also fit for B_0 in vivo? (Figure 4 doesn't show any fit, but the results in figure 3 look like they could include B_0 fitting, or am I missing something?) I think the PLOF approach is assuming a relaxivity (rather than a ratio of exchange rates), which allows it to fit for both concentration and exchange rate when using only a single irradiation power and duration, but I'm not certain. Why the different exchange-rate constraints (in PLOF vs neural network)? Related to this, why does the PLOF fitting have systematic errors? Is it due to finding a local minimum? (It is unclear in the "Numerical simulation" section if only a single guess is used in the least squares search.) If so, then a more global fit should be tried to make a fair comparison. Most importantly, if there is a systematic bias in the least squares fitting using the Bloch equations, possible reasons should be discussed.

Minor points:

1. Why “Online Methods” instead of “Methods”?
2. “Numerical Simulation” section refers to figure 2, but I think you mean some of the subfigures in figure 1.
3. Figure 1e does the B0 map in “Hz”. Did you mean ppm? Also figures 4f and 4h should use the same units, if possible.

No.: NCOMMS-19-24815

Responses to the comments of reviewers

The authors greatly appreciate the comments and suggestions from the reviewers, incorporation of which has improved the manuscript. In the following, the original comments of the referees are written in *blue italic* font and our response in regular black font.

Reviewer #1

It's well known that ^{31}P MRS offers quantitative insight into the metabolic processes underpinning muscle function and its pathology. Although resting-muscle measurements of pH, Pi, PCr etc can be informative, the most useful measurements in this regard are time-series of pH and PCr during exercise and recovery protocols. ^{31}P MRS has well-known limitations of hardware, signal-to-noise and spatiotemporal resolution (well described in this paper's Introduction) which have limited its clinical and indeed also research application. This paper describes the extremely thorough optimization and validation of a novel and ingenious approach to PCr measurements in skeletal muscle implementable on clinical scanners, which offers a partial way round these limitations. In metabolic terms, the main thing missing, compared to ^{31}P MRS, is pH measurement. The other main limitation is temporal resolution, and I would suggest that this is the next thing to improve.

Against this background, I make only two suggestions about the manuscript:

1. I suggest that a very brief summary of the points I have made in relation to the background and possible use of this technique would usefully set the scene for the less specialist reader.

Indeed, the capability to measure the combined effects of pH and PCr concentration changes is the most important feature of ^{31}P NMR. We added a brief summary of the abovementioned points to the Introduction and Discussion sections for better understanding of our work:

Introduction (Page 3): *In addition to PCr measurement, ^{31}P MRS also provides information about pH, inorganic phosphate and adenosine phosphates (ATP, ADP, AMP) in tissue. In practice, ^{31}P MRS is most commonly applied to monitor the time dependencies of pH and PCr variation during exercise and recovery for assessing mitochondrial function.^{7,8}*

Discussion (Page 10): *This is important for the future clinical applications of PCr mapping, where ANNCEST has the potential to address the major target of mapping of the time dependencies of PCr concentration during exercise and recovery. A preliminary result of estimating spatially resolved map of PCr recovery rate constant using ANNCEST is shown in Supplementary Materials Section 5 and a recovery time constant of 70.7 ± 55.4 s was obtained, which is consistent with that reported in the previous study (63.1 ± 25.9 s)⁴³. However, the temporal resolution of PCr ANNCEST in the current study (i.e. 90 s) was too low to capture very detailed*

dynamic changes. Future possibilities for reducing the scan time of PCr ANNCEST are adopting fewer saturation offsets or utilizing fast CEST sequences^{44, 45, 46}, which needs further study. Finally, as can be seen from Figs S1 and S2 in Supplementary Materials, the exchange rate depends on both pH and temperature. While we fit out the exchange rate, it is currently not trivial to separate out the effects of pH and temperature and this will be a topic for further study.

Reference

7. Kemp GJ, Meyerspeer M, Moser E. Absolute quantification of phosphorus metabolite concentrations in human muscle in vivo by ³¹P MRS: a quantitative review. *NMR in Biomedicine: An International Journal Devoted to the Development and Application of Magnetic Resonance In vivo* 20, 555-565 (2007).
8. Kemp G, Ahmad R, Nicolay K, Prompers J. Quantification of skeletal muscle mitochondrial function by ³¹P magnetic resonance spectroscopy techniques: a quantitative review. *Acta physiologica* 213, 107-144 (2015).
43. Schmid AI, et al. Dynamic PCr and pH imaging of human calf muscles during exercise and recovery using (³¹P) gradient-Echo MRI at 7 Tesla. *Magn Reson Med* 75, 2324-2331 (2016).
44. Zhang Y, et al. Chemical exchange saturation transfer (CEST) imaging with fast variably-accelerated sensitivity encoding (vSENSE). *Magn Reson Med* 77, 2225-2238 (2017).
45. Heo H-Y, Zhang Y, Lee D-H, Jiang S, Zhao X, Zhou J. Accelerating chemical exchange saturation transfer (CEST) MRI by combining compressed sensing and sensitivity encoding techniques. *Magn Reson Med* 77, 779-786 (2017).
46. Zhang S, Liu Z, Grant A, Keupp J, Lenkinski RE, Vinogradov E. Balanced Steady-State Free Precession (bSSFP) from an effective field perspective: Application to the detection of chemical exchange (bSSFPX). *J Magn Reson* 275, 55-67 (2017).

2. Obviously this would be marginal, given the current time resolution, but is it in any way feasible that the ANNCEST timecourse data in Fig 5a (and for comparison the ³¹P MRSI data in Fig 5b), specifically the recovery-from-exercise timepoints, could be used to calculate a spatially resolved map of PCr recovery rate constant? Such a thing would be interesting and physiologically relevant, being straightforwardly interpretable in terms of spatial distribution of mitochondrial function, and is surely a major way this technique could and should be developed. If (as I suspect) the data are too sparse, temporally, to support even a proof-of-principle calculation at this stage, I think it would be worth saying how this would work and why it would be useful. The point is, of course, that such kinetic measurements probe muscle energy metabolism and its abnormalities in a more effective (in some sense e.g. (sensitive, interpretable) way than resting [PCr] measurements.

Following the reviewer's suggestion, we demonstrate the feasibility of calculating a spatially resolved map of

the PCr recovery time constant using PCr ANNCEST. The PCr ANNCEST data shown in Fig.5 were adopted. To compensate for the sparsity of sampling time points, the baseline data was inserted behind the last recovery data as an additional sample. The time of first recovery data was set to 0 and the time interval of PCr mapping was 90 s. The PCr recovery rate constant τ_{PCr} was fitted using the following equation (Parasoglou P, et al. Magn Reson Med 2012;68(6):1738-1746.):

$$PCr(t) = PCr_0 + \Delta PCr \cdot (1 - e^{-t/\tau_{PCr}}) \quad (\text{SEq2})$$

where PCr_0 is the PCr concentration at the end of exercise, ΔPCr refers to the difference in PCr concentration between resting and end of exercise. The fitting was accomplished by the MATLAB built-in function “lsqcurvefit”. The fitting ranges of PCr_0 , ΔPCr and τ_{PCr} were set to 1~ 50 mM, 1~ 50 mM and 1 ~ 300 s, respectively.

From the typical fitting result shown in Fig. S5 (b), PCr concentrations of gastrocnemius muscles show exponential recovery after exercise and the fitted τ_{PCr} 72.6 s is consistent with that obtained by ^{31}P MR (63.1 ± 25.9 s) (Schmid AI, et al. Magn Reson Med 2016;75(6):2324-2331.). However, except for the gastrocnemius muscles, the fast PCr recovery of other muscles (see Figure 5b) could not be well captured and fitted due to the coarse temporal resolution (Fig. S5c). Improving the temporal resolution of PCr ANNCEST may yield a more detailed and robust τ_{PCr} map, which needs further study. The above results were added to the Supplemental Material S5.

Figure S5. Proof-of-principle calculation of the PCr recovery time constant τ_{PCr} . (a) T₂ weighted anatomical image and selected ROI (red circle). (b) The fitting results of the selected ROI indicated in (a). (c) Fitted τ_{PCr} map corresponding to PCr ANNCEST data shown in Fig. 5. The fitted τ_{PCr} values within the regions circumscribed by the white line are 70.7 ± 55.4 s (mean \pm s.d.).

Reviewer #2

The authors presented a pipeline that utilizes artificial neural networks (ANN) to quantify Phosphocreatine (PCr) concentration with high immunity to MRI interferences including magnetic field (B_0) and radio-frequency transmission (B_1) inhomogeneities. The methods have been tested on both phantoms and

exercised skeletal muscle. The study documents the ability of the ANNN -based chemical exchange saturation transfer (ANNCEST) for measuring PCr and diagnosing related diseases.

Strength

1. The paper introduces a framework for an important application that has a great interest to researchers for medical image analysis society, especially those pursuing in-vivo PCr quantification.

2. The study is well-designed and validated.

3. Most of the literature work references is up-to-date.

4. The paper is well- written, and results are nicely discussed.

We thank reviewer for positive comments and summarizing the strengths of our work.

Weaknesses

5. The core computational algorithm used in the pipeline (ANN) is already published, which is not a problem in itself, but the rationale behind the specific choice should be discussed. Additionally, the scientific contribution of the manuscript, however, has been compensated for by the application itself and method evaluation.

As the reviewer mentions, ANN has been well established and successfully applied to many diverse areas nowadays. The initial idea of this study and the choice of ANN were inspired by the work of Bo Zhu et al. (Nature 2018;555(7697):487-492), who demonstrated that conventional image reconstruction methods can be replaced by ANN-based method with much-improved performance. Similar to image reconstruction, CEST quantification is a kind of inverse problem, where the useful information (e.g. concentration and exchange rate) needs to be decoded from the acquired data (i.e. the intensities in a Z-spectrum). The encoding process of CEST MRI can be well described by the Bloch-McConnell equations and Z-spectra can be easily generated with known parameters. However, due to the complexity of the Bloch-McConnell equations, an accurate solution is hard to derive especially with the presence of possible B_0 and B_1 effects, which means decoding quantitative concentrations and exchange rates from Z-spectra is challenging. Here, ANNCEST provides a new dimension for CEST quantification. A fully connected feed-forward neural network is known to be good at finding patterns behind one-dimensional data (Goodfellow I, et al. Deep learning: MIT press; 2016. Schmidhuber J. et al. Neural Networks 2015; 61:85-117.), which is inherently suitable for the CEST quantification. Instead of needing to derive the solution of the Bloch-McConnell equations, we train the ANN with a large annotated dataset to extract the relationship between the Z-spectrum and quantifiable parameters. Even though ANNCEST is a powerful tool, careful design of the acquired Z-spectrum and quantifiable parameters is still required, as described in the Discussion section. We have added the abovementioned statements in the revised manuscript (Page 8) to clarify the specific choice of ANN.

6. The online Methods is misplaced after discussion. How many is the training data? Also, 80-15-5 split is the author's choice or a standard procedure? Although the authors adapt the ANN, they still need to add details about their method, optimization, hyper-parameter tuning, etc.

Similar to the reviewer, we prefer the Methods before the Results, but we need to follow the mandatory format of Nature Communications, with manuscripts organized in the order: Abstract -> Introduction -> Results -> Discussion -> Methods.

The number of Z-spectra used for neural network training was 10^5 . We have clarified this in the revised manuscript. The 80-15-5 split is our choice and the default split provided by MATLAB is 70-15-15. To improve the neural network generalization and avoid overfitting, the training data were divided randomly into three sets (i.e. training set, validation set and test set). The training set was used for computing the gradient and updating the network weights and biases, and the error on validation set is monitored during the training process. During the initial phase of training, the error on validation set normally decreases. However, when the network begins to overfit the data, the error on validation set typically begins to rise. When this error increases for a specific number of iterations (40 in this study), the training of neural network is stopped, and the weights and biases at the minimum of the validation error are returned. The test set is designed to compare the performance of different ANN models, and while the error on test set is not used during training, in practice, it is still useful to monitor. If the error in the test set reaches a minimum at a significantly different iteration number than the error in the validation set, this might indicate a poor division of the data set. In this study, since the training Z-spectra were generated by randomizing the quantifiable parameters within certain ranges and no other ANN model was adopted, the test set is not critical for the training of neural network. Therefore, we reduced the portion of test set to 5% and increased the portion of training set to 80%. It can be seen from Fig. 1(c) that the 80-15-5 split works well, and small standard deviation is observed between repeated trainings of neural network. The details of neural network training, specified in the first paragraph of the Methods section (Page 11), have now been expanded.

7. One more limitation of the study is the data size. Power analysis should be conducted to determine the appropriate data size that can be used to draw the author's conclusions.

In this study, PCr mapping using ANNCEST was validated by comparison with ^{31}P 2D MRSI measures obtained before and during in-magnet plantar flexion exercise. Following the reviewer's suggestion, a power analysis was performed to determine the appropriate data size to draw the conclusion. Assume we accept a $p < 0.001$ as acceptable and a study with 95% power, the sample size for the study will be (Kadam P, Bhalerao S. Int J Ayurveda Res 2010;1(1):55-57.):

$$n = \frac{2 \times (3.2905 + 1.6449)^2 \sigma^2}{\Delta^2} \quad (\text{SEq3})$$

where σ refers to the estimated standard deviation and Δ indicates the difference in effect. In this study, we expected a 50% reduction in PCr concentration during exercise (i.e. $\Delta \approx 15$ mM), and the standard deviation of PCr concentrations is 7.84 mM based on the baseline data shown in Fig. 5 (c). According to Eq. SEq3, the required sample size is about 14. In this study, a pixel-by-pixel correlation analysis was performed to compare PCr maps obtained by ANNCEST and ^{31}P 2D MRS on resting and exercised human skeletal muscle. Each PCr map has $16 \times 16 = 256$ pixels, and the PCr maps of baseline, during holding, and 0.75 min of recovery were chosen. Even though only partial regions were chosen for correlation analysis, the effective data size from four subjects is 202, which is much larger than the required sample size. We added the power analysis in Supplementary Section Section 7.

8. Abstract does not contain any quantitative results. Also, please refrain from using abbreviations from abstract, unless you define them (see e.g., WHM)

Following reviewer's suggestion, we reduced the usage of abbreviations for B_0 and B_1 in the Abstract (we left the abundant PCr one) and also added some quantitative results:

Abstract (Page2): *The PCr ANNCEST outcomes strongly correlated with those from ^{31}P magnetic resonance spectroscopy ($R = 0.813$, $p < 0.001$).*

9. Quantitative results and statistical analysis have been conducted an included, however, comparison with other techniques is missing.

We have now added a description of how our method compares to ^{31}P measurements of pH and PCr concentration and compared the PCr recovery time from our experiments with the literature. Please see the responses to Reviewer 1 comment 1.1 for details.

Reviewer #3

The robust imaging of PCr on a 3T clinical system in 1.5 minutes is an exciting result that could have significant interest in the medical imaging field. While there are lots of interesting results, the logic and assumptions of the paper are not fully clear, with specific relevance to the benefit of the neural network approach vs the PLOF approach, lessening enthusiasm.

1. The overall logic of the paper is not fully clear, which is especially confusing since it is trying to combine what could be several papers (PLOF at 3T, ANN in steady-state vs non-steady state or phantoms vs in vivo, ANN validation with MRS, and ANN vs PLOF). It is divided into 5 parts: 1. ANN works on numerical 2-pool phantoms, and works better than PLOF/numerical fitting; 2. PCr generated signal contrast is maximized at a power and duration in vivo at 3T (which is different than that used in the digital or in vitro phantoms); 3. ANN works on phantoms at 3T, but with systematic spatial artifacts (which weren't

discussed). 4. ANN and PLOF work in muscle at 3T but ANN is more uniform and maybe better. 5. ANN results match MRS in exercised muscle. I'm not certain this is the division, which is part of the problem. The logical progression should be clearer. For example, the # of pools, experimental parameter ranges (including changing from steady state to non-steady state conditions), and fitting constraints change between the phantoms and the in vivo work. So what is being tested? Is it the general ANN approach? Wouldn't the validity of the approach depend on how valid the assumptions are, which is very different in vivo where, for example, the MTC parameters were unknown. The logic should be clearly stated, and the results and methods should match this logic and follow a clear progression.

We apologize for the confusion. This paper contains multiple steps, and all these steps serve one goal, i.e. developing high-quality PCr mapping using CEST for clinical practice at low field strengths. To achieve this goal, we developed and optimized PCr CEST experiment from data acquisition to data analysis to obtain optimal PCr CEST contrast and robust quantification result. For the data analysis, we proposed a novel CEST quantification framework dubbed ANNCEST, and its feasibility and efficiency were demonstrated on numerical simulation and phantom experiments (Figs. 1 & 2). For clinical data acquisition, we assigned the PCr CEST signal on human skeletal muscle at 3T and experimentally optimized the CEST contrast (Fig. 3). Combining the advanced CEST quantification method, ANNCEST, with the optimized PCr CEST acquisition, high-quality PCr mapping on human skeletal muscle was obtained (Fig. 4). As validation and application, we applied PCr ANNCEST to measure PCr concentration and its spatiotemporal changes during skeletal muscle exercise (Fig. 5). For better evaluating and validating our approach, Bloch equation fitting, PLOF, and ³¹P MRSI were carried out to show the performance of state-of-the-art methods. Following reviewer's suggestion, we added the following sections (in italic font) to Introduction and Discussion to make our paper clearer. To demonstrate the flexibility of ANNCEST in quantifying Z-spectrum with different parameters (e.g. number of pools, saturation length), we added Cr phantom results in Supplementary Materials Section 6.

Introduction (Page 4): *After first training and validating ANNCEST using numerical simulations and PCr phantom data at 3 Tesla, we optimize the PCr CEST acquisition to obtain maximum PCr contrast on human skeletal muscle and again train and apply ANNCEST. We then show the feasibility of applying ANNCEST to simultaneously quantify the PCr concentration of human skeletal muscle, the exchange rate of the guanidinium protons from PCr, and the B_0 and B_1 maps on a clinical 3T MRI scanner.*

Discussion (Page 7): *From the PCr phantom experiments shown in Fig. 2, due to the reduced CEST contrast, the exchange rate of 10 mM PCr phantom is more vulnerable to systematic imperfections and exhibits a larger standard deviation compared to the others. Therefore, optimizing CEST sequence to obtain maximum PCr contrast is critical for robust quantification since the PCr contrast in vivo is around 1% (Fig. 3 h & i).*

Discussion (Page 9): *The relatively homogeneous T_1 and MTC across the human skeletal muscle benefit the*

generation of training data^{35,36,37}. With optimal saturation parameters, training Z-spectra within a limited spectral range (1.3 ppm to 3.5 ppm) can be generated using a three-pool Bloch McConnell simulation, namely water protons, PCr guanidinium protons, and background. The background including the contributions from MTC and all other metabolites can be well represented by a single pool (Fig. S4e in Supplementary Materials). The discernible PCr guanidinium CEST peak in vivo provides a unique opportunity for ANNCEST to learn the relationships between Z-spectrum and PCr concentration, exchange rate, B_0 and B_1 (Supplementary Section S3&4), and to apply the learned knowledge to simultaneously quantify these multiple parameters, as demonstrated in Figs. 4 & 5.

Reference:

35. McDaniel JD, et al. Magnetization transfer imaging of skeletal muscle in autosomal recessive limb girdle muscular dystrophy. *Journal of computer assisted tomography* 23, 609-614 (1999).
36. Marty B, Carlier PG. Physiological and pathological skeletal muscle T1 changes quantified using a fast inversion-recovery radial NMR imaging sequence. *Sci Rep* 9, 6852 (2019).
37. Varghese J, et al. Rapid assessment of quantitative T1, T2 and T2* in lower extremity muscles in response to maximal treadmill exercise. *NMR Biomed* 28, 998-1008 (2015).

Supplementary Materials (Section 4):

Since the MTC is homogeneous across human skeletal muscle (McDaniel JD, et al. *Journal of computer assisted tomography* 1999;23(4):609-614.) and no other metabolites show indiscernible CEST peak on Z-spectrum, the background of Z-spectrum on human skeletal muscle can be regarded as an additional pool and well presented by Bloch-McConnell equations, as shown in the newly added Fig. S4e.

Figure S4e. The MTC dominant background can be well presented by a pool with a mean concentration of 8 M and mean exchange rate of 30 Hz.

Supplementary Materials (Section 6):

For the training of ANNCEST, we didn't make any assumptions. We feed the neural network with training Z-spectra and quantifiable parameters, and the other parameters, such as T_1 , T_2 , MTC, number of pools and saturation length, are blind for the neural network. The key thing we want to test in this paper is that if ANN can learn the relationship between desired parameters (e.g. concentration and exchange rate) and Z-spectrum and apply the learned knowledge to quantify new Z-spectrum. It can be seen from the Supplementary Materials (Fig. S4) that the depth, width and offset of CEST peak are related to concentration, exchange rate, and B_0 introduced frequency shift, respectively, and these effects can be fully exploited by ANN and applied to simultaneously quantify these parameters, as demonstrated by the results in this paper. The flowchart of ANNCEST was added in Fig. S6-1. In order to demonstrate that ANNCEST is still valid with a different number of pools and saturation length (non-steady-state), Cr phantom experiments were performed and the results are given in Fig. S6-2&3.

ANNCEST is a data-driven quantification method, which is designed to extract relevant features from Z-spectra and utilize them to create a predictive tool based on the pattern hidden inside. The flowchart of ANNCEST is shown in Fig. S6-1.

Figure S6-1. Schematic flowchart of ANNCEST. The annotated Z-spectra are simulated by Bloch-McConnell equations with proper assumptions and parameters that mimic realistic tissue conditions. ANNCEST is trained to extract relevant features between Z-spectrum and quantifiable parameters. During neural network training, only Z-spectra and quantifiable parameters are provided, and the other parameters (e.g. T_1 , T_2 , MTC, and noise) are blind for the neural network.

In order to demonstrate that ANNCEST is valid for other metabolites with a different number of pools and saturation length, Cr phantom experiments were performed. The training data for these experiments were

generated using the Bloch-McConnell equations. The frequency offsets of the Z-spectra ranged from 0.5 to 4 ppm with a total offset number of 50. The offsets of Cr CEST peaks were set to 1.95 ppm. The T_1 and T_2 of water protons were set to 2.6 s and 1.8 s, respectively, according to the measurements on the phantom. The saturation power and duration were 0.6 μ T and 3 s. The concentration, exchange rate, and B_0 inhomogeneity were chosen randomly from the ranges of 5 to 105 mM, 100 to 350 Hz, -0.2 to 0.2 ppm, respectively. Gaussian white noise with zero mean value and 0.0015 standard deviation was imposed on the simulated Z-spectra. The number of Z-spectra used for neural network training was 105. The training results are shown in Fig. S6-2, which reflects that ANNCEST can find strong correlations between Z-spectra and quantifiable parameters. For validation, we applied the trained ANNCEST to quantify Z-spectra of Cr phantom obtained at room temperature (25 $^{\circ}$ C), and the results are given in Fig. S6-3. An excellent correlation ($R=0.9996$) was observed between the ground truth and predicted phantom Cr concentration. The related Bland-Altman analysis of concentration is shown in Fig. S6-3f. The exchange rate obtained by ANNCEST (237.8 ± 17.6 Hz) was consistent with that from the previous study (239~301 Hz at 25 $^{\circ}$ C, pH 6.9-7.0) (Goerke S, et al. NMR Biomed 2014;27(5):507-518.). The predicted B_0 map (Fig. S6-3e) showed a strong correlation (0.9851) with that obtained by water saturation shift referencing (WASSR) MRI, as illustrated in Fig. S6-3h.

Figure S6-2. Error histogram and regression plot of the neural network training results for Cr phantom experiment.

Figure S6-3. Typical Validation of ANNCEST at 3T (preclinical MRI) on a phantom consisting of test tubes with different concentrations of Cr. All experiments were performed at room temperature (25°C). (a) The arrangement of the Cr phantoms with different concentrations. (b) Representative Z-spectra extracted from one pixel of each of the Cr phantoms. The ANNCEST-predicted concentration (c), exchange rate at 1.95 ppm (d) and B_0 maps (e) for the CEST experiments collected using a 3 s saturation pulse of 0.6 μ T. (f) Bland-Altman plot for the predicted concentration and ground truth. (g) The exchange rate quantification results. The bar and error bar indicate the mean value and standard deviation across each phantom, respectively. (h) Bland-Altman plot for the predicted B_0 map and referenced B_0 map obtained via WASSR method.

2. Were all simulations, training, and measurements (except for a separate exchange rate measure) done at 3T? The legend in figure 1b indicates a 300Hz variation, which would match the stated 0.4 ppm at 17.6T, not 3T. If the phantom work was done at 17.6 T, I don't understand the logic.

All the simulations, training and measurements shown in the manuscript were performed at 3T. The 300 Hz in Fig. 1b refers to the exchange rate instead of B_0 . We have added labels for concentration, exchange rate and B_0 values, in Fig. 1b to clarify this now.

3. *What are the assumptions and constraints, and what drives these decisions? From my reading, assumptions in the numerical training of the phantoms include 2 pools, the ratio of exchange rates at 2 and 2.5 ppm (only good at a single temperature and pH?), $B_1 = 0.6 \mu\text{T}$ with no inhomogeneity, and water T_1 and T_2 values. The in vivo training makes very different choices, and many of the parameters are not listed (e.g. MTC, T_2 and lineshape). In both cases, it is unclear what assumptions are made about the PCr T_1 and T_2 , but they must be in the coupled Bloch equations. The training assumes a single T_1 , but doesn't that change with, for example, exercise? Discussion of possible problems and systematic errors is necessary, along with discussion of why these constraints were chosen (e.g. is it impossible to do the fitting without specifying the ratio of exchange rates?).*

For the training of ANNCEST, we didn't make any assumptions. We feed the neural network with training Z-spectra and quantifiable parameters, and the other parameters, such as T_1 , T_2 , MTC, number of pools and saturation length, are blind for the neural network. The key thing we want to test in this paper is that if ANN can learn the relationship between desired parameters (e.g. concentration and exchange rate) and Z-spectrum and apply the learned knowledge to quantify new Z-spectrum.

The assumptions we made are for the generation of training data. In this paper, the training Z-spectra were generated by Bloch-McConnell equations and the possible combinations of Z-spectra increase exponentially with the number of variables. Therefore, reducing the number of variables according to specific applications is a compromised but feasible solution at current stage. The T_1 and T_2 values of phantom and human skeletal muscle were obtained by additional measurements. The MTC-dominant background is homogeneous across human skeletal muscle (McDaniel JD, et al. Journal of computer assisted tomography 1999;23(4):609-614.) and can be well presented by an additional pool with parameters shown in Supplementary Materials Fig. S4e. The ratio of exchange rates for PCr two peaks was determined by the NMR experiments with respect to temperature and pH as shown in Supplementary Materials Fig. S2, and the variation of ratios is around 0.51 with different pH and temperature. According to previous study (Varghese J, et al. NMR Biomed 2015;28(8):998-1008.), the T_1 will increase around 10% after exercise, and we didn't

include T_1 change in the training data of human skeletal muscle, which means the PCr mapping after exercise may contain the contamination from T_1 . In further study, better design of training data is required to improve the accuracy of ANNCEST. Following reviewer's suggestion, we added related discussion in the revised manuscript.

Discussion (Page 9): Previous studies^{2, 37, 38} have shown that the pH, T_1 and T_2 can change after exercise. We included the effects of pH and T_2 on training data by adopting varying exchange rates and T_2 values. However, we didn't consider T_1 change (around 10% increment) in the current study, which means the PCr mapping after exercise may contain some contamination from T_1 . Further efforts are required to build more accurate training data and hence facilitate the accuracy of ANNCEST.

Reference

2. Arnold DL, Matthews PM, Radda GK. Metabolic recovery after exercise and the assessment of mitochondrial function in vivo in human skeletal muscle by means of ^{31}P NMR. *Magn Reson Med* 1, 307-315 (1984).

37. Varghese J, et al. Rapid assessment of quantitative T_1 , T_2 and T_2^* in lower extremity muscles in response to maximal treadmill exercise. *NMR Biomed* 28, 998-1008 (2015).

38. Schmid AI, et al. Exercising calf muscle T_2^* changes correlate with pH, PCr recovery and maximum oxidative phosphorylation. *NMR Biomed* 27, 553-560 (2014).

Figure S2. Temperature (b) and pH (c) dependence of the exchange rates for the PCr CEST peaks at 2.6 ppm and 1.95 ppm, respectively. The exchange rate ratio between 2.6 ppm and 1.95 ppm peaks as functions of temperature (d) and pH (e) are also plotted, respectively. The temperature dependence experiments were performed with a pH value of 7.3, while the pH dependence experiments were performed at 37 °C.

4. Is there a fair comparison to PLOF approach? Are the assumptions the same? Does PLOF also fit for B_0 in vivo? (Figure 4 doesn't show any fit, but the results in figure 3 look like they could include B_0 fitting, or am I missing something?) I think the PLOF approach is assuming a relaxivity (rather than a ratio of exchange rates), which allows it to fit for both concentration and exchange rate when using only a single irradiation power and duration, but I'm not certain. Why the different exchange-rate constraints (in PLOF vs neural network)? Related to this, why does the PLOF fitting have systematic errors? Is it due to finding a local minimum? (It is unclear in the "Numerical simulation" section if only a single guess is used in the least squares search.) If so, then a more global fit should be tried to make a fair comparison. Most importantly, if there is a systematic bias in the least squares fitting using the Bloch equations, possible reasons should be discussed.

The procedures of PLOF method are: (1) Z-spectrum excluded CEST peak is used to fit the background; (2) fit the whole Z-spectrum with fixed background and get the true apparent relaxation rate of CEST peak. The offset of CEST is flexible during the fitting, which possesses some resistance against B_0 inhomogeneities. The assumption of PLOF method is that the background of Z-spectrum is broad and smooth, which can be presented by a polynomial function. This assumption may break down when including the Z-spectrum close to water, which has higher curvature compared to that further from water. Therefore, we discarded the saturation offsets 1.3 ~ 1.6 ppm during PLOF fitting. From the fitting results shown in Fig. 3, the 1.6 ~ 2.1 ppm is enough for the background fitting. In our opinion, the systematic errors for PLOF come from the following factors: First, the true apparent relaxation rate ($T_{1\rho}$ based) is affected by B_1 , so an additional B_1 map is needed to correct for local effect changes induced by B_1 inhomogeneity. Second, the inevitable noise will degrade the fidelity of PLOF method, as well as Bloch equation fitting and other fitting based methods. From the demonstrations shown in the below Figure, satisfactory consistency was obtained by PLOF with different boundaries in the case without noise. However, when adding noise to Z-spectrum, the quantification results oscillated with different boundaries, which meant that the fitting method could not find a unique solution and the quantification results depended strongly on the fitting parameters.

Figure for reviewer 3. A typical representation of PLOF results with different noise levels and boundaries for CEST peak. The noise added in (d ~f) is Gaussian white noise with zero mean value and 0.0015 standard deviations.

Minor points:

5. Why “Online Methods” instead of “Methods”?

We removed “Online” in the revised manuscript to avoid confusion.

6. “Numerical Simulation” section refers to figure 2, but I think you mean some of the subfigures in figure 1.

We thank reviewer for pointing out this typo. We replaced “Figure 2” with “Figures 1(d-f)” in the revised manuscript.

7. Figure 1e does the B₀ map in “Hz”. Did you mean ppm? Also figures 4f and 4h should use the same units, if possible.

Fig. 1e indicates the exchange rate map in “Hz” and Fig. 1f exhibits the B₀ map in “ppm”. Following the reviewer’s suggestion, we normalized the B₁ map in Fig. 4f using the reference power 0.6μT to make the units the same as Fig. 4h.

Reviewers' comments:

Reviewer #1 (Remarks to the Author):

Thank you - my comments have been satisfactorily addressed.

Reviewer #2 (Remarks to the Author):

The authors addressed all previous comments. I think the paper is ready for publication after proofreading

Reviewer #3 (Remarks to the Author):

The authors have addressed many of my concerns, but there are still a few method issues that need to be clarified.

1. The experiments as presented require a separate T1 and T2 measurement, and this requirement should be made clear. My reasoning is that the fitting is validated using training data that constrains T1 and T2 to the separately measured value. While the ANN is blind to this value, the overall experimental design and validation depends on using the correct (i.e. separately measured) values.
2. The parameters used for training and fitting are still not clear. For example, it is still not clear what T1 and T2 of the PCr protons (NOT water T1 and T2) were used in the training. Were they varied? Similarly, the methods section for the "Z-spectra for neural network training" for human muscle does not discuss MT/background. (Some info is given in figure S4e, but it shouldn't be that hard to piece things together.) I strongly suggest building off figure S6-1, perhaps by adding a corresponding table, that lists exactly the input parameters (and their ranges) for each of the experiments. This specification also relates to point #1, since if you can find a clear way to also specify the corresponding constraints, that would ideal. The goal should be to make it easy for others to reproduce.

No.: NCOMMS-19-24815A

Responses to the comments of reviewers

The authors again would like to thank the reviewers for taking time to provide constructive comments and helpful suggestions to improve the manuscript further. In the following, the original comments of the referees are written in *blue italic* font and our response in regular black font.

Reviewer #1

Thank you - my comments have been satisfactorily addressed.

Reviewer #2

The authors addressed all previous comments. I think the paper is ready for publication after proofreading

Reviewer #3

The authors have addressed many of my concerns, but there are still a few method issues that need to be clarified.

1. The experiments as presented require a separate T_1 and T_2 measurement, and this requirement should be made clear. My reasoning is that the fitting is validated using training data that constrains T_1 and T_2 to the separately measured value. While the ANN is blind to this value, the overall experimental design and validation depends on using the correct (i.e. separately measured) values.

Our newest data now indicate that T_1 and T_2 measurements are generally not needed. The T_1 and T_2 measurements in the original manuscript were useful for generating training data that mimic the real situation. Once the neural network is well trained, ANNCEST can simultaneously predict the concentration, exchange rate and B_0 with a simple input of Z-spectrum without the requirement for additional T_1 and T_2 measurements. We have now added additional simulations to show that the trained network is insensitive to T_1 and T_2 over the *in vivo* range (Marty B, Carlier PG. Sci Rep 2019;9(1):6852.) for exchange rate determination and only slightly sensitive to T_1 and T_2 for concentration determination. We have added related statements in the results section and discussion of the revised manuscript to clarify this:

Page 6-7: "In order to validate the robustness of ANNCEST against T_1 and T_2 variations, we applied trained ANNCEST to quantify Z-spectra simulated with different T_1 and T_2 values, and the results are given in Supplementary Materials Section 8. From Figure S8-1(b), when increasing T_1 over a large range from 1.0 s to 2.0 s, the quantified concentration increased only from 35.03 ± 1.95 mM to 37.33 ± 1.90 mM, in an approximately linear fashion. The exchange rate obtained by ANNCEST possessed excellent resistance against T_1 variation (160.7 ± 2.6 Hz at $T_1=1$ s vs 159.0 ± 3.3 Hz at $T_1=2$ s), as shown in Figure S8-1(c). The quantified concentrations and exchange rates as a function of T_2 are shown in Figures S8-1 (e,f). These results indicate that ANNCEST still can yield reasonable accuracy when T_2 varies from 15 to 50 ms."

Figure S8-1. Simulations showing that ANNCEST for PCr gives reasonable results over the relevant in vivo range of water T_1 and T_2 values. The gold standard concentration, exchange rate, B_0 , and B_1 were set to 35 mM, 160 Hz, 0 ppm and 0.6 μ T. The other parameters are listed in Supplementary Table S3-2. The simulations were repeated 50 times with the amount of noise added being varied. (a) Representative Z-spectra with water T_1 values ranging from 1.0 to 2.0s. The concentrations (b) and exchange rates (c) obtained by ANNCEST as a function of water T_1 . (d) Representative Z-spectra for water T_2 values ranging from 10 to 60 ms. The concentrations (e) and exchange rates (f) obtained by ANNCEST as a function of T_2 . The solid line refers to the mean value of 50 repetitions and the light-colored area represents the standard deviation.

2. The parameters used for training and fitting are still not clear. For example, it is still not clear what T_1 and T_2 of the PCr protons (NOT water T_1 and T_2) were used in the training. Were they varied? Similarly, the methods section for the “Z-spectra for neural network training” for human muscle does not discuss MT/background. (Some info is given in figure S4e, but it shouldn’t be that hard to piece things together.) I strongly suggest building off figure S6-1, perhaps by adding a corresponding table, that lists exactly the input parameters (and their ranges) for each of the experiments. This specification also relates to point #1, since if you can find a clear way to also specify the corresponding constraints, that would ideal. The goal should be to make it easy for others to reproduce.

The T_1 and T_2 values for PCr protons were set to 0.05 s and 0.02 s, respectively. We have added this detail in the revised manuscript (Page 12). Also, we added new data to indicate the robustness of ANNCEST against T_1 and T_2 variations of PCr proton, as shown in Supplementary Materials Figure S8-2.

Supplementary Materials Section 8: The robustness of ANNCEST against T_1 and T_2 variations of PCr proton is shown in Figure S8-2. From the results, the concentration and exchange rate obtained by ANNCEST possesses excellent resistance against T_1 variation of PCr proton (34.75 ± 2.35 mM and 163.7 ± 3.5 Hz at 30 ms v.s. 36.06 ± 2.16 mM and 158.6 ± 4.9 Hz at 70 ms). ANNCEST still can yield satisfactory accuracy when PCr proton T_2 varies from 15 ms to 25 ms, e.g. concentration varies between 34.9 mM and 36.9 mM, while the exchange rate is in the range of 162.8 Hz to 159.0 Hz.

Figure S8-2. Simulations showing that ANNCEST for PCr gives reasonable results over a range of T_1 and T_2 values for PCr protons. The gold standard concentration, exchange rate, B_0 , and B_1 were set to 35 mM, 160 Hz, 0 ppm and $0.6 \mu T$. The other parameters are listed in Supplementary Table S3-2. The simulations were repeated 50 times with the amount of noise added being varied. (a) Representative Z-spectra with water T_1 values ranging from 30 to 70 ms. The concentrations (b) and exchange rates (c) obtained by ANNCEST as a function of T_1 of the PCr protons. (d) Representative Z-spectra with PCr proton T_2 values ranging from 10 to 30 ms. The concentrations (e) and exchange rates (f) obtained by ANNCEST as a function of PCr proton T_2 . The solid line refers to the mean value of 50 repetitions and the light-colored area represents the standard deviation.

Following the reviewer's suggestion, we added two tables in the Supplementary Materials Section 3 to clarify the parameters and their ranges. We also added a discussion for the MTC/background fitting in the Methods section (Page 13):

Z-spectra for neural network training (Page 13): The MTC-dominated background signal was incorporated as an additional pool with a concentration of 8M for the exchanging protons and an

exchange rate of 30 Hz. The T_1 and T_2 values for the background signal were set to 1 s and 9.1×10^{-6} s, respectively. The lineshape for the background signal was a Super Lorentzian function. The goodness of background fitting using the above parameters is shown in Supplementary Materials Fig. S4(e).

Supplementary Table S3-1: The parameters used to generate training data for PCr phantom

Saturation power	0.6 μ T
Saturation length	10 s
Z-spectrum range	0.5 - 4 ppm
T_{1w}	2.6 s
T_{2w}	1.8 s
$T_{1(\text{PCr protons})}$	0.05 s
$T_{2(\text{PCr protons})}$	0.02 s
CEST peaks	1.95 ppm & 2.5 ppm
Exchange rate ratio	1.95 ppm: 2.5 ppm = 1 : 2.19
Exchange rate range	50 - 200 Hz
Concentration ratio	1.95 ppm: 2.5 ppm = 1 : 2
Concentration range	5 - 85 mM
B_0 inhomogeneity range	-0.4 - 0.4 ppm
Noise	Gaussian white noise with zero mean value and 0.0015 standard deviations

Supplementary Table S3-2: The parameters used to generate training data for human skeletal muscle

Saturation power	0.6 μ T
Saturation length	800 ms
Z-spectrum range	1.3 - 3.5 ppm
T_{1w}	1.2 s
T_{2w}	15 - 35 ms
$T_{1(\text{PCr protons})}$	0.05 s
$T_{2(\text{PCr protons})}$	0.02 s
CEST peak	2.5 ppm
Exchange rate range	80 - 230 Hz
Concentration range	0 - 100 mM
B_0 inhomogeneity range	-0.25 - 0.25 ppm
B_1 inhomogeneity range	0.5 - 0.7 μ T
MTC/background concentration	8 M
MTC/background exchange rate	30 Hz
MTC/background T_1	1 s
MTC/background T_2	9.1×10^{-6} s
MTC/background lineshape	SuperLorentzian
Noise	Gaussian white noise with zero mean value and 0.0035 standard deviations

REVIEWERS' COMMENTS:

Reviewer #3 (Remarks to the Author):

The authors have addressed my comments.